# Frontal and parietal planning signals encode adapted motor commands when learning to control a brain–computer interface

Enrico Ferrea[1]*, Pierre Morel[1,2], Alexander Gail[1,3,4,5]*

1 German Primate Center, Göttingen, Germany, 2 Univ. Littoral Côte d'Opale, Univ. Artois, Univ. Lille, ULR 7369 - URePSSS - Unité de Recherche Pluridisciplinaire Sport Santé Société, Calais, France, 3 Faculty of Biology and Psychology, University of Göttingen, Göttingen, Germany, 4 Bernstein Center for Computational Neuroscience Göttingen, Göttingen, Germany, 5 Leibniz ScienceCampus Primate Cognition, Göttingen, Germany

* eferrea@dpz.eu (EF); agail@dpz.eu (AG)

## Abstract

Perturbing visual feedback is a powerful tool for studying visuomotor adaptation. However, unperturbed proprioceptive signals in common paradigms inherently co-varies with physical movements and causes incongruency with the visual input. This can create challenges when interpreting underlying neurophysiological mechanisms. We employed a brain–computer interface (BCI) in rhesus monkeys to investigate spatial encoding in frontal and parietal areas during a 3D visuomotor rotation task where only visual feedback was movement-contingent. We found that both brain regions better reflected the adapted motor commands than the perturbed visual feedback during movement preparation and execution. This adaptive response was observed in both local and remote neurons, even when they did not directly contribute to the BCI input signals. The transfer of adaptive changes in planning activity to corresponding movement corrections was stronger in the frontal than in the parietal cortex. Our results suggest an integrated large-scale visuomotor adaptation mechanism in a motor-reference frame spanning across frontoparietal cortices.

## Introduction

Intracortical brain–computer interfaces (BCIs) have shown success in restoring lost motor functions in individuals with tetraplegia and chronic stroke, primarily by extracting signals from the motor cortex [1–6]. Recent advancements have extended this approach to signals obtained from the parietal cortex [7–10]. Besides their translational applications, BCI paradigms provide a crucial tool for studying the neurophysiological basis of motor learning [11–23]. Unlike conventional motor learning paradigms involving physical limb movement, BCI control establishes an experimentally controllable mapping from brain activity to effector motion through the neural decoder.

**Data availability statement:** All data reported in the figures are available through the Supporting information provided with this article.

**Funding:** German Research Foundation (DFG, Germany, Grant Number FOR-1847-GA1475-B2), received by AG. German Research Foundation (DFG, Germany, Grant Number SFB-889), received by AG. Federal Ministry for Education and Research (BMBF, Germany, Grant Number 01GQ1005C), received by AG. Federal Ministry for Education and Research (BMBF, Germany, Grant Numbers 01GQ0814), received by AG. Link to founding agencies: https://www.dfg.de/en https://www.bmftr.bund.de/EN/Education/HigherEducation/Funding/funding_node.html. The funders had no role in study design, data collection and analysis, decision to publish, or preparation of the manuscript.

**Competing interests:** I have read the journal's policy and the authors of this manuscript have the following competing interests: AG is a member of PLOS Biology's editorial board.

**Abbreviations:** AI, alignment indexes; BCI, brain–computer interface; FP, fronto-parietal; FMAs, floating microelectrode arrays; KF, Kalman filter; M1, primary motor cortex;MC, manual control; MIP, medial intraparietal area;PC, principal component; PD, preferred direction; PMd, dorsal premotor cortex;PRR, parietal reach region; RMS, root-mean-square; VMR, visuomotor rotation;VR, virtual reality.

Additionally, BCI paradigms help isolate the impact of visual feedback on movement adaptation by keeping proprioceptive feedback constant during visuomotor perturbations [16]. This approach reduces potential confounds inherent in the adaptation of physical movements, which co-vary with proprioceptive feedback, and emphasizes the visual aspect of feedback control.

In our study, we employ a BCI visuomotor rotation paradigm in rhesus monkeys to explore the role of visual feedback and the contributions of parietal and premotor sensorimotor areas to BCI adaptation. We ask whether both brain regions (i) contribute to adaptation irrespective of an immediate impact on the produced motor behavior, (ii) reflect the same or different sensory or motor-related spatial dimensions of fast visuomotor adaptation, and (iii) show adaptation of motor planning activity consistent with motor behavior.

In BCI learning, behavioral performance can be improved by exploiting an existing repertoire of neural dynamic states ("within-manifold" learning) or by reconfiguring the neural network structure ("outside-manifold" learning). In response to perturbations of a BCI decoder, short-term learning does not appear to significantly affect the correlation patterns among neurons in the motor cortex [15,20,24]. This suggests that at least the local network structure supplying input to the decoder (i.e., the controlling motor area) remains largely unchanged and constrains neural adaptation during short-term learning. However, with longer-term learning, the brain can be trained to generate new activity patterns in the controlling units that deviate from the expected correlation pattern of the network. Such outside-manifold learning aligns with the view that the tuning of individual neurons can be adapted independently during BCI learning [13,24–27].

A way how the decoder input could be adapted without changing the network structure is by means of "re-association". According to this view, a specific neural activity pattern associated with compensating for the introduced visuomotor perturbation is recruited from the range of patterns available within the existing network structure [15,18–21]. This view implies changes in activity, but under the constraint of an unchanged network structure also for neighboring neurons, such that also non-controlling units in the controlling area that do not provide input to the decoder adapt their activity in a manner consistent with changes in the BCI controlling units. Based on these previous findings, we expect that the comparatively fast adaptation to visuomotor rotation, achievable within a single session of a few hundred trials, is accompanied by such re-association learning. After establishing that this is indeed the case in our visuomotor adaptation task, we will test three implications of this hypothesis.

First, we will address the question of whether changes in neural activity during BCI learning occur consistently not only in nearby noncontrolling neurons, as shown previously [13,18,28], but also across a larger-scale sensorimotor network, including frontoparietal circuits. Previous studies in humans and monkeys have not extensively examined the extent of adaptation within a distributed neural network, including regions that are not directly involved in generating or updating motor commands but may still play a crucial role in sensorimotor integration. These regions, such as the parietal cortex, are expected to contribute to motor learning [29,30]. In a human BCI

motor learning study involving the prefrontal cortex, dorsal premotor cortex (PMd), primary motor and sensory cortices, and the posterior parietal cortex, neural changes occurred with long-term learning and were associated nonspecifically with reduced cognitive demand [31]. In contrast, we aim to understand the neural characteristics of fast adaptation, which is particularly desirable for BCI learning and, in conventional visuomotor rotation learning, is often associated with the updating of an internal model [32,33]. Fast adaptation is especially important for BCI applications because it enables users to achieve accurate and efficient control with minimal training time, which is critical for practical and clinical use. We compare the contributions of noncontrolling neurons residing locally next to the BCI controlling neurons in the frontal areas or remotely in parietal areas to test if fast adaptation effects generalize to remote brain regions.

Second, we ask whether premotor and parietal brain regions not only both adapt but do so in the same spatial frame of reference. Specifically, we test if neural changes reflect the adapted motor command (this is mandatory for the controlling units, but unclear for noncontrolling units) or if the parietal cortex rather reflects the associated adapted sensory feedback. Preparatory activity in frontoparietal motor planning areas includes information about the reach goal represented in different body-related frames of reference. While encoding at the level of neural populations and often also at the level of the individual neurons shows typically mixed representations of different reference frames, different coding schemes predominate across areas. For example, the parietal reach region (PRR) in the medial intraparietal area (MIP) predominantly encodes reach targets in an eye-centered reference frame [34], whereas area 5d in the posterior parietal cortex represents reach goals more strongly in a hand-centered frame [35]. PRR representations can relate both to the visual location and the physical goal of the movement [36,37]. Nearby sensorimotor areas V6A in the anterior wall of the parieto-occipital sulcus and PEc at its shoulder show gradually more selectivity for the hand position, additionally to target and gaze-related information [38–40]. Similarly, the PMd encodes reach targets in a mixed reference frame, integrating the relative positions of the hand, eye, and target [41,42]. Thus, while both premotor and parietal regions are involved in reach planning and show overlapping tuning properties during instructed delay, they differ with respect to their neuroanatomical embedding. Posterior parietal networks integrate sensory information from multiple modalities, including visual input, and contain motor-goal information during reach planning without directly driving motor output [43,44]. Premotor cortex, on the other hand, has direct connections to primary motor cortex (M1) and its more caudal parts even to spinal cord [45]. Here, we test if these differences between frontal and parietal motor planning areas reflect in differences in the predominant frame of reference when motor command and sensory feedback are dissociated during sensorimotor adaptation.

During a visual working memory task, frontoparietal areas can even contain spatial information not related to the body, but relative to a visual object, which is an allocentric form of spatial representation [46]. The possibility of such spatial coding in a reach context, which does not reflect the planned physical arm movement, raises the question in which spatial frame of reference frontoparietal cortex represents reach-associated information during BCI learning and whether parietal and frontal areas share a common frame of reference. Rather than reflecting trial-to-trial updates of the corrected movement to produce adapted motor outputs, as observed in the motor and premotor cortices [11,28,47,48], parietal dynamics could reflect the unchanged sensory state of the system during the planning period before movement onset, as part of its role in state estimation [49–55].

Third, we propose that motor adaptation is reflected in changes in motor signals during movement preparation already and that planning activities co-vary with signals during movement control. Fast adaptation at the neural level has been previously described from a dynamical system perspective [22,23,56–58]. According to this view, the neural state, which represents the activity pattern across the population of neurons in motor areas, follows established dynamics to generate movements [59–63]. During movement preparation, the neural state is initialized with different conditions for different movements and then evolves based on the network-inherent dynamics during movement execution. This concept is particularly applicable to ballistic reaches toward spatially segregated targets in space. The fixed neural dynamics are determined by the stable recurrent connectivity of the network itself [19,61,64]. According to this view, adaptation could

                                                                                          

be implemented by updating of the initial states. Hence, adaptation induced neural changes should reflect in late planning activity, at a time when uncertainty about the position of a BCI-controlled cursor position deteriorates planning signals [65].

Over the course of fast adaptation, the idea, neural dynamics are initialized with changing conditions to compensate for movement errors from trial to trial, while the network structure in the motor cortex remains stable. A BCI learning study in monkeys indicated that such adaptive state initialization probably applies to BCI controlling units in the motor cortex [23]. In line with this finding, other work suggested that motor learning involves systematic changes in preparatory activity within the motor cortex to adapt to a force field [66]. It is yet unclear to what extent such adaptation of initial states also applies to changes in preparatory activity in the larger frontoparietal network, including noncontrolling units and areas outside the motor cortex. By introducing an instructed delay in the BCI movement task, we can compare adaptation of the preparatory states with the movement-associated dynamics. We ask whether such a relationship between the two states exists also for visuomotor adaptation and whether the relationship changes over the course of learning.

In this study, we conducted a 3D visuomotor rotation task under BCI control to investigate neural adaptation across the distributed frontoparietal network, including M1, PMd, and PRR. We selectively connected only a subset of units, referred to as controlling units, to the decoder to assess whether neural signatures of adaptation extend beyond the units immediately controlling the BCI movements. After demonstrating visuomotor rotation adaptation in BCI control of 3D movements and its compatibility with the idea of fast within-manifold learning, we address our three main research questions. First, we test if adaptation is distributed and also affects the coding in parietal areas, remote from the frontal lobe areas that controlled the BCI movements. Second, we determine the spatial frame of reference in which the noncontrolling units co-vary with the controlling units that determine the behavior. Third, we test the hypothesis that re-association learning leads to adaptation of movement planning signals consistent with the observed adaptation in motor behavior.

## Results

### Monkeys learned to control movements in 3D virtual reality with BCI

To better understand the neural mechanisms underlying movement planning and execution, we designed an experiment to investigate adaptive changes in neural dynamics during a memory-guided reach task under BCI control. Two rhesus monkeys (Y and Z) performed a memory-guided 3D center-out reach task using a computer cursor in a virtual reality (VR) environment (Figs 1A and S1A). The task involved the presentation of a target, followed by a memory period during which the target was no longer visible. The monkeys were then required to move the cursor to the previously cued target in a corner of a 3D cube (Fig 1B). This delay period between target presentation and movement execution allowed us to investigate adaptive changes in neural dynamics during movement planning, independent of immediate sensory feedback, and compare these to the adapted motor commands during execution.

The monkeys controlled the cursor either through natural hand movements (manual control, MC) or via a BCI driven by intracortical neural activity (Fig 1C). During MC trials, the cursor movements in virtual space were aligned with the animals' hand movements in physical space. In BCI trials, the monkeys kept their unrestrained hand in a resting position while controlling the cursor mentally. S1B Fig shows that residual hand movements were minimal and barely correlated with cursor speed (Pearson's correlation: monkey Y, $r = 0.085$, $p < 0.001$, $n$ (total number of decoded speeds) = 575,258; monkey Z, $r = 0.072$, $p < 0.001$, $n = 108,933$). This means the animal managed to control the cursor movement without any longer attempting own physical movements, neither during planning nor during movement, which ensured a constant somatosensory input during BCI trials. During baseline trials, i.e., without visuomotor perturbation, monkey Y successfully drove the cursor to the target in 99% of the movement trials (trials in which the animal successfully completed the planning period without, e.g., fixation breaks; see Materials and methods) under MC and in 89% of the BCI trials. Monkey Z succeeded in 97% of the movement trials under MC and in 85% of the BCI trials. Only successful trials were analyzed.

We recorded brain activity from three regions: the primary M1, PMd, and PRR. Monkey Y had 64 electrodes in each of M1, PMd, and PRR, while monkey Z had 64 electrodes in M1 and 96 electrodes in each of PMd and PRR (S1C Fig). Units

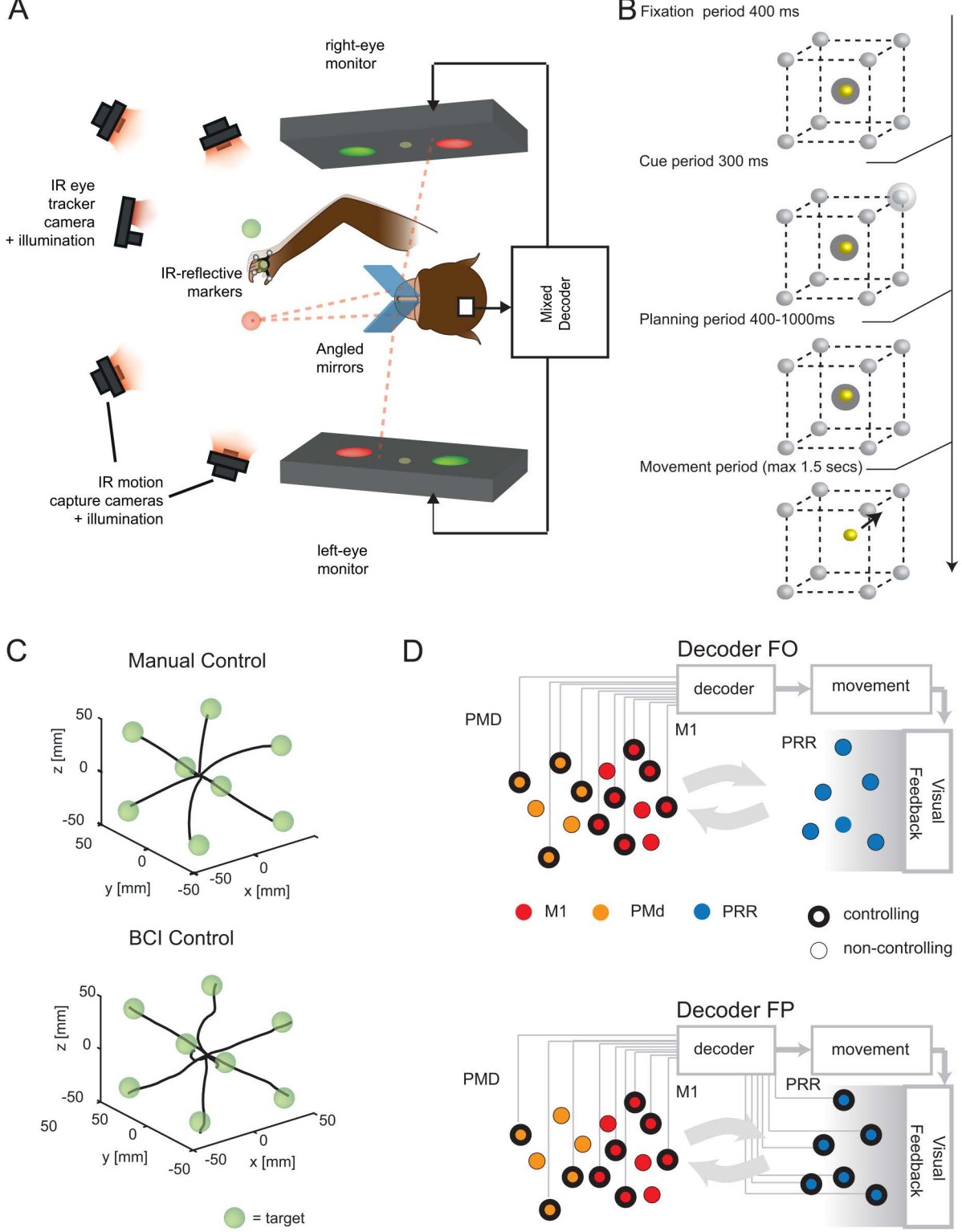

**Fig 1. Experimental setup and decoding scheme for 3D virtual reality (VR) task. (A)** Schematic representation of the 3D VR setup allowing control of movements through manual control (MC) and brain–computer interface (BCI). The setup includes four infrared cameras for online tracking of hand position using reflective markers, enabling realistic 3D movements and decoder calibration for the BCI task. The monkey's other arm was

gently restrained, and gaze position was monitored using an infrared eye tracker. The scheme was drawn manually by the authors. **(B)** Memory-guided center-out reach task in 3D. In each trial, one of the eight corners of the 3D cube was briefly cued as the target and had to be reached after a variable memory period. **(C)** Averaged 3D trajectories obtained from one experimental session in Monkey Y during 3D reaches performed under the BCI condition. **(D)** Two alternative decoding schemes. A subset of M1 and PMd cells was always involved in decoding, represented by black circle outlines. In contrast, PRR cells were either fully integrated into the decoding process (decoder FP) or entirely disconnected (decoder FO).

from all three areas were selectively connected (controlling units) or not connected (noncontrolling units) to the decoder (Fig 1D). This design allowed us to investigate whether adaptive mechanisms in individual neurons depended on their immediate causal influence on movement, which is only guaranteed for connected units. Subsets of PMd and M1 units were always involved in the decoder, whereas PRR units were either entirely connected (in combination with PMd-M1; Decoder fronto-parietal (FP), Fig 1D, lower) or completely disconnected (Decoder frontal-only (FO), Fig 1D, upper). This setup enabled us to study adaptive mechanisms in PRR even without its direct involvement in generating movements and to compare it to the case when PRR directly contributes to the control of BCI movements.

## Adaptation of BCI movements was induced by perturbed visual feedback in a 3D visuomotor rotation task

We utilized a short-term visuomotor rotation (VMR) adaptation paradigm tailored to our 3D experimental setup [67] to induce repeated BCI learning (Fig 2). In the daily sessions, first, the decoder was recalibrated until animals became proficient in controlling unperturbed cursor movements without computer support (see Materials and methods). We then recorded 160 baseline trials before we introduced the perturbation, a rotation of 30° to the visual feedback (cursor) in the fronto-parallel X–Y plane, for 300 trials (Fig 2A). In each daily session, the perturbation was consistently either clockwise or counterclockwise only (S1 Table). The perturbation was applied globally throughout the workspace, which means it affected movements to all eight targets but only in the X and Y dimensions, not in the depth dimension away from the body (Fig 2A, left). Finally, the perturbation was removed during the washout phase, which continued until the animals disengaged from the task (Fig 2A, right). In designing this study, we leveraged previous findings from [67], which showed that human subjects adapted most easily to perturbations applied in the fronto-parallel plane.

To visualize movement adaptation, we projected the 3D movement trajectories of the controlled cursor into the X–Y plane affected by the perturbation (perturbation plane). During the baseline phase, the trajectories were on average relatively straight toward the target. During early rotation trials, as expected as a consequence of the perturbed feedback, trajectories in the perturbation plane curved in the direction of the applied feedback perturbation (Fig 2A, right). The average trajectories were less curved in late rotation trials. In the early washout phase, the trajectories on average curved in the opposite direction compared to during perturbation. The reduced curvature in late rotation trials and inverse curvature observed during washout indicate successful adaptation, which we quantified jointly across all target directions. To obtain averaged trajectories across targets, we rotated them for each target such that the X-axis aligned with the center-to-target direction and the Y-axis aligned orthogonally to this (perturbation direction, Fig 2B; see Materials and methods). Fig 2C depicts the average cursor trajectories across all eight targets and for all sessions for monkey Y. Additionally, the corresponding "motor" signal, representing the trajectories that would be produced by the unperturbed decoder using the neural activity of the controlling units, is visualized (Fig 2B–2C).

To quantify adaptation, we calculated the trial-by-trial angular movement error ($a$) from the starting direction of the movement. To have a good enough estimate of movement direction while keeping the influence of online movement corrections small, we measured the angle of the cursor position at the halfway point of the trajectory relative to the straight line connecting the starting position and the target position (Fig 2C).

On average, during baseline trials, the angular error was close to zero and did not decrease over time, as the fitted data with exponential decay did not show significant differences from zero (Monkey Y: intercept = 0.637, $p = 0.26$, decay = 0.006, $p = 0.43$, $n = 51$; Monkey Z: intercept = 1.5, $p = 0.26$, decay = 0.0029, $p = 0.8$, $n = 15$). It increased to about 20° at

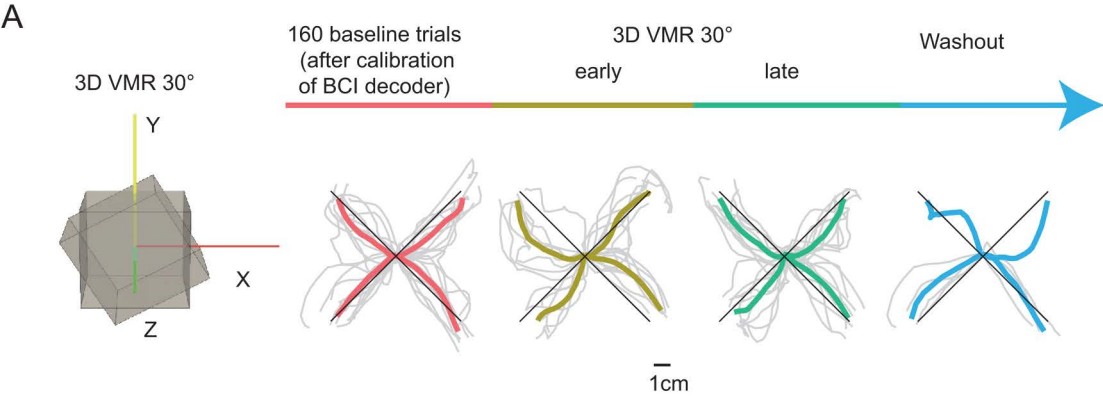

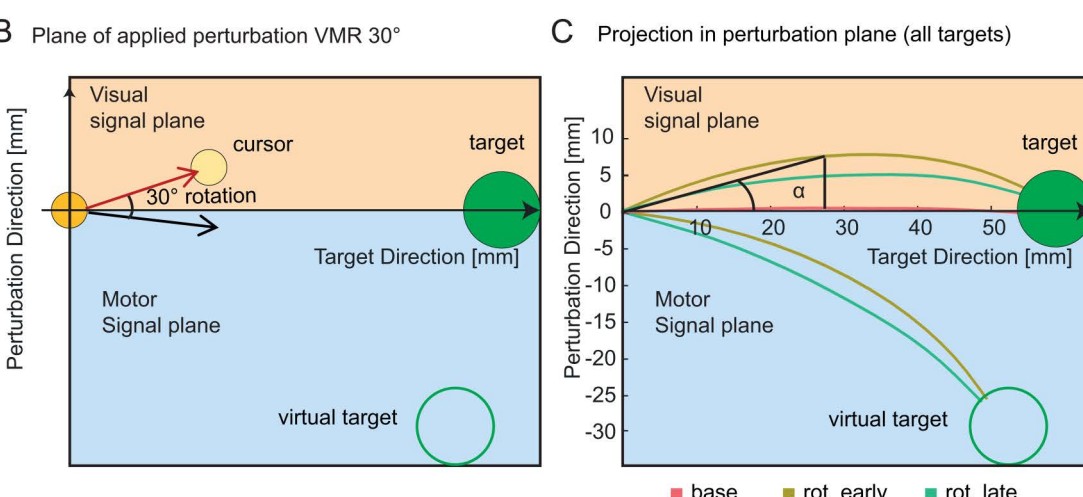

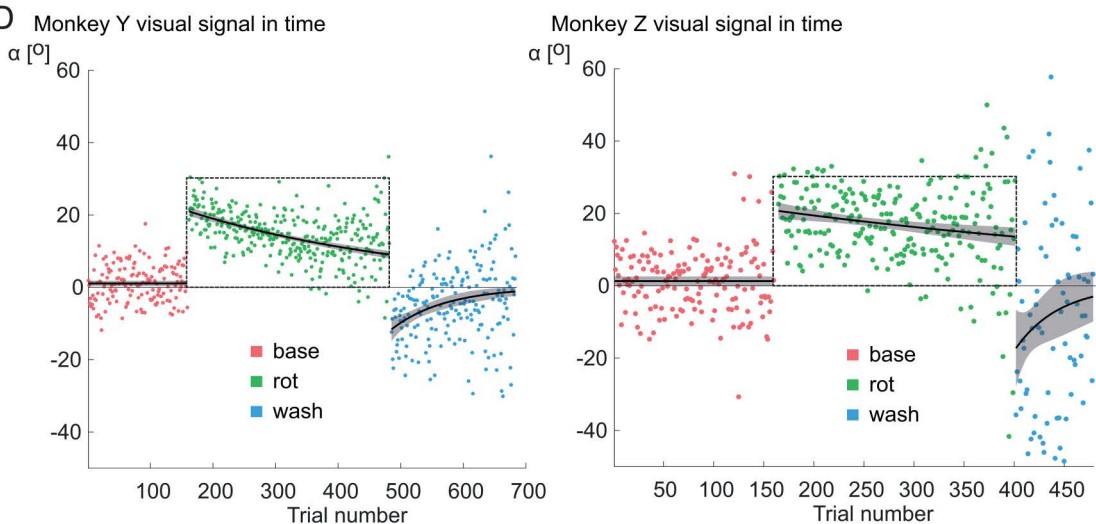

**Fig 2. Experimental paradigm and BCI adaptation. (A)** Left: visual representation of 3D-VMR. In different experimental sessions, the 3D movements were rotated 30° clockwise or counterclockwise in the frontal-parallel plane. The Z-axis points towards the monkey's head and was unperturbed. Right: the experimental protocol consisted of a baseline phase followed by a 30° VMR and a washout phase where the visual perturbation was removed. Below each phase of the experimental progression, example traces from a single experimental session in Monkey Y are shown. **(B)** In the plane of the applied

perturbation, deviations above the zero line correspond to positions in the visual signal plane. Conversely, negative deflections in trajectories exist in the motor signal plane and correspond to the motor commands issued to counteract the visual perturbation. For averaging across all target directions, the trajectories were rotated into a shared reference frame. One axis was aligned with the perturbed direction, while the other was aligned with the vector from the center to the target. In this plane, trajectories above the zero line represent positions in the visual signal plane, while trajectories below zero pertain to the motor signal plane. Additionally, we calculated the angle at the midpoint of each trajectory as the angle between the perturbation direction and the vector from the center to the target. **(C)** For the analysis, the 3D trajectories were projected onto the 2D plane of the applied perturbation. The trajectories showed stereotypical adaptation profiles during rotation and negative aftereffects during washout. **(D)** The angle at the midpoint of the trajectory, plotted as a function of trial number, reveals stereotypical adaptation profiles in response to the introduced VMR. The perturbation angle approaches 30° upon perturbation introduction, gradually decreases with learning, and then deviates in the opposite direction during the washout phase following perturbation removal. Numerical data are available in S1 Data.

the onset of the VMR perturbation and gradually decreased as the monkeys adapted their neural activity to regain better control (Fig 2D). Fitting the data with an exponential decaying function during the perturbation phase indicated significant visuomotor adaptation (Monkey Y: $n = 51$, intercept = 20.99, $p < 0.001$, decay = −0.0026, $p < 0.001$; Monkey Z: $n = 15$, intercept = 20.69, $p < 0.001$, decay = −0.0018, $p = 0.0017$).

During the washout phase, movement errors were committed in the opposite direction to the applied perturbation, consistent with a negative aftereffect as typically seen in visuomotor rotation adaptation. Fitting the data with an exponential decay revealed a significant intercept for both animals (Monkey Y: intercept = −11.63, $p < 0.001$, $n = 26$; Monkey Z: intercept = −17.255, $p = 0.0018$, $n = 4$), indicating significant aftereffects for both animals. These findings suggest a change in the sensorimotor transformation achieved during BCI adaptation trials.

## Preserved covariance structure suggests within-manifold learning during BCI-VMR

In BCI-controlled movements, behavioral changes represented by the cursor on the screen (i.e., the decoder output) directly mirror the activity of the neural population that controls the decoder. When faced with a VMR-type perturbation, this controlling neural group can restore its performance by re-associating certain target directions with different movement directions [15,18]. The re-association hypothesis means that the neurons can direct the cursor to the desired target using activity patterns consistent with those observed during the center-out reach task on which they were originally trained, associating a direction suited to compensate for the perturbing rotation. In contrast to the re-configuration hypothesis, the covariance structure between different neurons should change very little with re-associations, since the network remains stable. This stability implies that the neural dynamics that define the low-dimensional neural subspace (i.e., the neural manifold), which explains most of the neural variance before adaptation (during baseline), also explain most of the variance during perturbation and after adaptation (washout).

To test the re-association versus re-configuration hypothesis, we calculated the alignment index [62] of the neural manifolds between the baseline trials and the other experimental phases (S2 Fig). Alignment should be high in the case of re-association. The alignment index is computed as the ratio of the explained variance when projecting the neural activity from the other experimental phases onto the principal components (PCs) derived from the baseline activity (see Materials and methods section). We focused on the first four PCs, which capture the majority of the variance in the baseline trials (S3A, S3C, S3E, and S3G Fig). Supporting the re-association hypothesis, we find that a substantial portion of the initially explained variance remains constant throughout the learning process. In fact, the average difference between the alignment indices during late rotation and the cross-validated alignment index during baseline is smaller than 1% for all the different neural populations that we calculated (S2 Fig). This suggests a preservation of the manifold structure during VMR adaptation in the frontal motor network.

## Noncontrolling units in M1-PMd encode the same corrective task parameters during adaptation of BCI movement as controlling units

Our primary research goal was to identify the spatial reference frames of movement parameters encoded by neurons without a direct causal link to the movement (noncontrolling units) in frontal versus parietal brain areas. During BCI

movements, noncontrolling units could encode movement parameters in either a visual coordinate system (reflecting the cursor movement) or a physical coordinate system (reflecting intended limb movements). To distinguish between these spatial encoding strategies, we used an offline decoding approach to reconstruct movement trajectories from noncontrolling units.

First, a Kalman filter (KF) decoder (as for online decoding) was trained offline with neural activity of the noncontrolling units during BCI baseline trials (training set, Fig 3A) to reproduce the cursor trajectories actually produced by the animal with the controlling units during baseline. During unperturbed baseline trials, these trajectories represent both the motor command (of the controlling units) and visual cursor feedback alike, since both are spatially congruent (identical except for sensory noise).

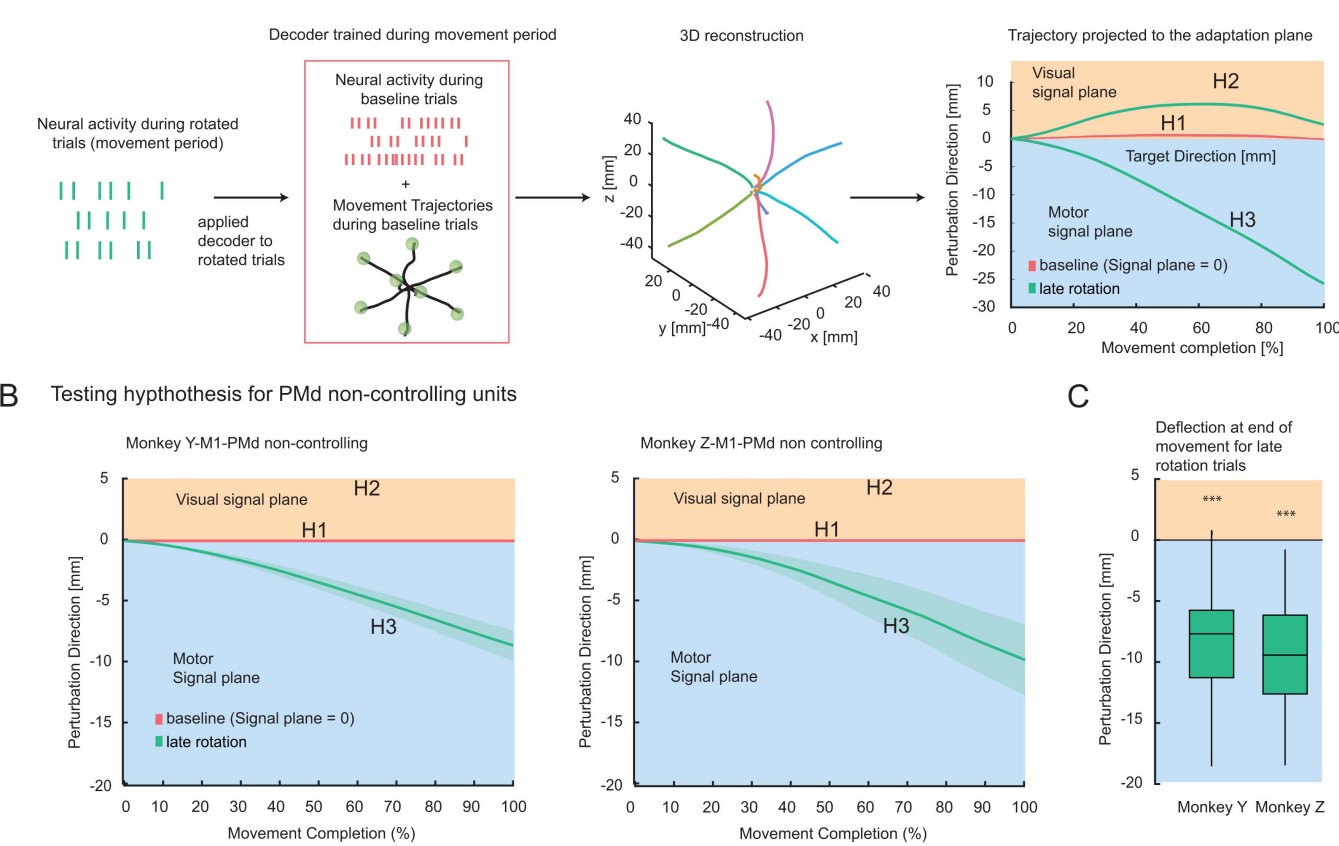

**Fig 3. Offline decoder approach. (A)** Schematic illustration of the offline decoding principle during the movement period. A velocity Kalman filter decoder was applied to reconstruct theoretical memory trajectories from neural activity (first column). During the movement period, real BCI trajectories were used for regression with neural firing rates from noncontrolling neurons during baseline trials (second column, training set). After calibration during baseline trials, the decoder was used to reconstruct continuous trajectories during perturbation trials (third column). The example averaged trajectories demonstrate continuous movements toward the target. In the last column, the reconstructed 3D trajectory from the offline decoder is projected onto the perturbation plane for perturbation trials. These projections assess visual-like or motor-like encoding, where positive deviations along this axis correspond to the visual signal, and negative deflections indicate motor output during adaptation. **(B)** Reconstruction and projection of the offline trajectories from M1-PMd in the adaptive dimension for the late adaptation phase. The depicted trajectories represent the averages across all experiments where nondecoding units were collected. The traces display the average (solid color) and the bootstrapped confidence interval (shaded colors) for baseline (pink) and late rotation (last 50% of the trials per session) trajectories (green). **(C)** Box plot showing deflections of the trajectories at the end of movement for both animals during late rotation trials. A significant negative deflection indicates motor-like encoding (*$p < 0.05$, ***$p < 0.001$). The whiskers represent the 5th and 95th percentiles. Numerical data are available in S2 Data.

Second, this decoder was then applied to the neural activity during VMR trials of the same neural population (test set, Fig 3A). During VMR trials, the motor command of the controlling units and the visual cursor feedback are spatially incongruent. This allows us to test which spatial reference frame the noncontrolling units better correspond to. More specifically, the projection of the decoded trajectories along the perturbation axis (deflection measure) allowed us to study whether neural activity of the noncontrolling units during adaptation reflects (i) a stationary, nonadaptive signal (H1), (ii) the adapting visual cursor trajectories (H2), or (iii) the adapting motor signal as produced by the controlling units (H3). Straight trajectories to the target as during baseline would reflect a nonadaptive signal in noncontrolling populations, corresponding to a lack of deviation in the perturbation direction (Fig 3A). Deviation from baseline deflecting in the positive direction of the perturbation axis would reflect the visual signal (visual signal plane; corresponding to perturbation direction greater than 0), while deflection in the negative direction would reflect the motor command (motor signal space; corresponding to perturbation direction <0).

In our results, the negatively deflected reconstructed trajectories in both animals show that noncontrolling units in M1 and PMd reflect the motor command similarly to controlling units (Fig 3B–3C). To accurately capture this deflection, we used a trajectory deflection measure in our analysis. This approach differs from the angular movement error ($\alpha$) that we previously calculated to quantify adaptation, where $\alpha$ was measured at the midpoint of the trajectory relative to the straight line connecting the starting and target positions (Fig 2C). We chose the deflection measure for the reconstructed trajectories because it accounts for the overall shape and direction of the entire movement path.

The mean deflection of the activity of monkey Y at the end of the movement (100%) was −8.69 mm ± 4.43 mm (mean ± standard deviation), significantly different from zero (one-sample $t$ test, $t(50)$ = −14.010, $p < 0.001$). For monkey Z, the mean deflection of activity at the end of the movement was also significantly below zero, at −9.88 mm ± 5.27 mm ($t(14)$ = −7.250, $p < 0.001$). To isolate area-specific adaptation between M1 and PMd, we also tracked how each neuron's preferred direction (PD) evolved during learning. This analysis revealed robust, systematic PD rotations that were independently measurable in both M1 and PMd (S4 Fig). These tuning shifts are well-aligned to partially compensate for the experimentally induced visual rotation.

To gain deeper insight into the level of adaptation achieved by the noncontrolling neuronal population, we calculated the angle representing the deviation in the perturbation direction at the final level of adaptation. This angle was measured between the reconstructed trajectory of each population and the direct path to the target. To compare the adaptation levels across different populations, we then scaled the final angle reached by each neural subpopulation relative to that of the controlling unit population and called this the "relative gain." The relative gain quantifies the deviation along the perturbation axis relative to the level of adaptation achieved. By expressing these deviations as a fraction of the total adaptation level exhibited by the controlling units, we could assess and compare how much each population contributed to adapting to the perturbation. Specifically, the M1-PMd noncontrolling population reached a relative gain of 59% for Monkey Y and 94% for Monkey Z, underscoring the significant adaptation observed in the M1-PMd noncontrolling population.

## Controlling and noncontrolling units in PRR share the same spatial frame of reference during adaptation of BCI movement as controlling units in M1-PMd

We next investigated how VMR adaptation affects PRR activity, focusing on whether PRR's adaptation-associated neural dynamics share the same spatial frame of reference as the frontal regions. By analyzing the adaptation process in PRR both with direct influence on movement control (decoder FP, controlling PRR units) and without it (decoder FO, noncontrolling PRR units), we determined whether PRR predominantly mirrors visual feedback or plays an active role similar to frontal areas. The overall neural yield in PRR was lower than in the combined M1-PMd recordings for both subjects. To still achieve similar baseline performance across conditions and to have enough noncontrolling units in PRR for analysis, the controlling PRR neurons were combined with a subset of M1-PMd units in the FP decoder sessions. This design choice was motivated by the fact that we observed lower performance in pre-testing trials when using PRR alone, possibly

due to its weaker motor coding properties. Still, this experimental strategy allowed us to compare adaptation of PRR while partially causing the movement or not causing it directly at all.

First, we evaluated whether the respective contributions of M1-PMd and PRR to the cursor movement were balanced during FP decoding, or rather predominated by M1-PMd inputs. Each cell's contribution was determined by multiplying the norm of the vector that translates decoded speed into its firing rate summed with its baseline firing rate (Fig 4A), a similar approach as described in [68]. The distributions of the contribution values differed only modestly between PRR and M1-PMd (Cohen's $d$: $d = 0.67$ for Monkey Y, $d = 0.38$ for Monkey Z). This suggests that both the PRR and M1-PMd areas similarly drive the motor output of the decoder. We also ensured that after baseline trials, neither of these areas significantly decreased their firing rate during adaptation which might indicate a substantial silencing of the area. Average firing rates between baseline and adaptation did not differ significantly (Mann–Whitney test, $p > 0.05$; Fig 4B). Together, these results indicate an active contribution of PRR to decoder adaptation.

Changes in PRR activity in response to the VMR exhibited motor-like characteristics similar to M1-PMd (Fig 4C). Across both decoder configurations (FP and FO), significant deflections from zero were observed at the end of movement. For monkey Y, under FP decoding, the mean deflection was $-1.64 \pm 1.56$ mm ($t(10) = -3.48$, $p = 0.006$); under FO, $-1.85 \pm 2.25$ mm ($t(39) = -5.20$, $p < 0.001$). For monkey Z, under FP decoding, deflection was $-10.90 \pm 2.87$ mm ($t(4) = -8.49$, $p = 0.001$); under FO, $-5.18 \pm 5.08$ mm ($t(4) = -12.37$, $p = 0.0001$). These results indicate a motor-like encoding during adaptation, qualitatively similar to M1-PMd (Fig 4C). Even when PRR was not contributing to the control of the

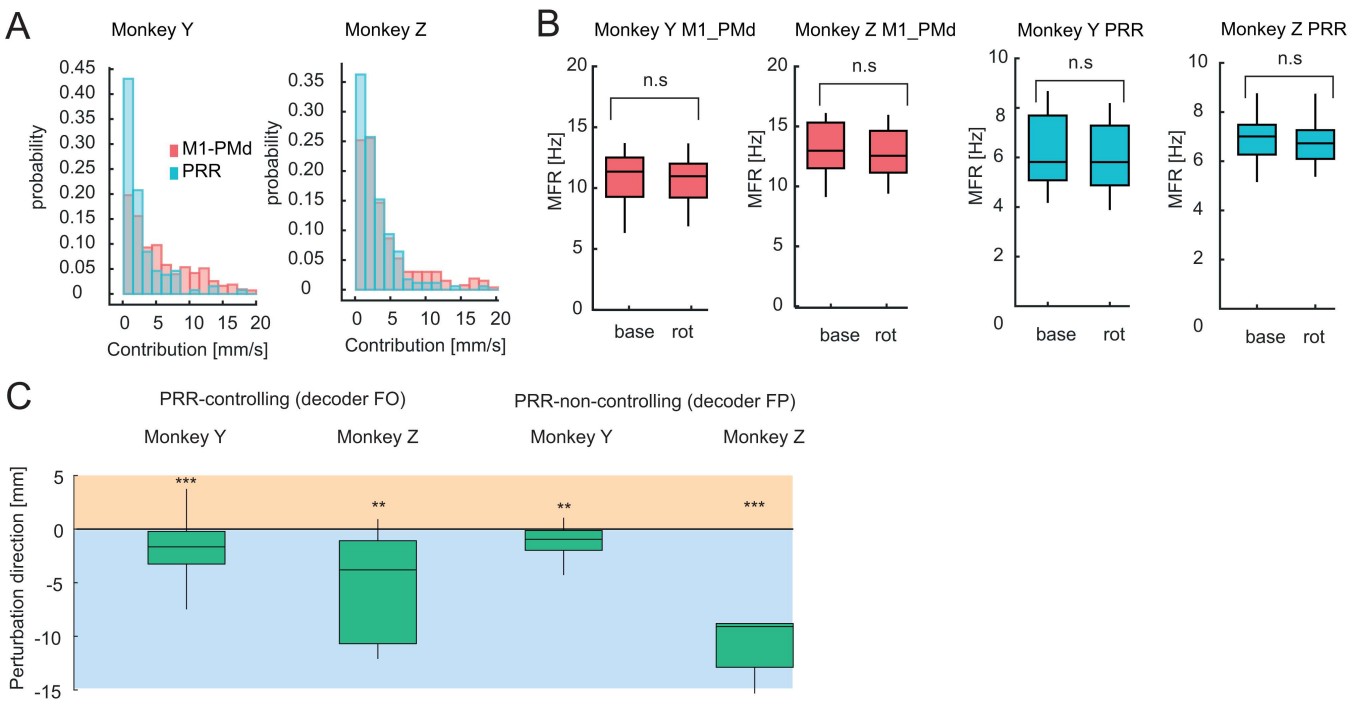

**Fig 4. PRR controlling and noncontrolling adaptation during movement execution. (A)** Distributions of the norm of the linear regression vectors from velocity to neural firing rate for every single unit contributing to the decoder, color-coded depending on the neural population. Numerical data are available in S3 Data. **(B)** Mean firing rates for PRR and M1-PMd controlling units during baseline and late rotation epochs show that neither PRR nor M1-PMd is reducing their firing rate when compared to baseline (all $p$ values nonsignificant with Mann–Whitney test, $n = 11$ for Monkey Y, $n = 5$ for Monkey **Z**). Numerical data are available in S3 Data. **(C)** Similar to Fig 3C, the bar plots show the deflection of the trajectories at the end of the reach. Each distribution consists of all PRR neurons and is tested against zero for the two different decoding schemes and the two monkeys separately (**$p < 0.01$, ***$p < 0.001$). Numerical data are available in S2 Data.

decoder, an equivalent pattern of adaptation was observed, indicating motor-related changes rather than representation of the visual feedback. For the PRR controlling units, Monkey Y showed a 13% relative gain to the final adaptation, while Monkey Z showed 64%. When PRR was noncontrolling, Monkey Y showed a 14% relative gain, and Monkey Z achieved 37%.

In summary, our findings suggest that PRR exhibits substantial motor-like encoding during movement adaptation, even when it is not directly controlling the movement, highlighting its involvement in the adaptation process. This indicates that PRR plays a more integral role in motor adaptation than merely reflecting visual feedback.

## Comparing movement planning and execution to uncover neural adaptation dynamics in BCI learning

An important aspect of our study design is to compare movement planning and execution to understand the dynamics of neural adaptation. Specifically, we ask whether motor preparation and the associated initial states of neural dynamics are updated trial-by-trial during adaptation, or if only online motor control changes during BCI learning. The transition between planning and movement phase was comparable in the BCI and MC contexts. We applied a PC decomposition and cross-projected the neural activity during planning onto the dimensions that best explained the activity during movement. We found that the explained variance in the planning activity, when projected into the first four dimensions of the movement subspace, was significantly reduced in comparison to the movement activity (S3 Fig, paired $t$ test, all comparisons $p$ < 0.001. Monkey Y, MC: M1-PMd $mean_{plan}$ = 0.46, $mean_{mov}$ = 0.86, $t(49)$; PRR $mean_{plan}$ = 0.87, $mean_{mov}$ = 0.91, $t(49)$. BC: M1-PMd $mean_{plan}$ = 0.67, $mean_{mov}$ = 0.92, $t(49)$; PRR $mean_{plan}$ = 0.88, $mean_{mov}$ = 0.96, $t(49)$. Monkey Z, MC: M1-PMd $mean_{plan}$ = 0.53, $mean_{mov}$ = 0.76, $t(14)$; PRR $mean_{plan}$ = 0.63, $mean_{mov}$ = 0.74, $t(14)$. BC: M1-PMd $mean_{plan}$ = 0.62, $mean_{mov}$ = 0.83, $t(14)$; PRR $mean_{plan}$ = 0.79, $mean_{mov}$ = 0.87, $t(14)$). The subspace misalignment was more pronounced in frontal areas compared to parietal areas in both monkeys. The alignment patterns were equivalent in both manual and BCI trials. These results show substantial differences between the manifold structure in the planning and movement phases not only in MC but also in BCI. We attribute these change in neural state to a transitions from planning while not attempting cursor movements during the delay period to actually producing a cursor move towards the target during the movement period. If the animals had attempted cursor movements during the delay period of the BCI trials already, the release of the cursor blockade at the end of the delay period would have led to an immediate cursor movement, since neural response modulations directly affect cursor movements in BCI control. Instead, what we observed in BCI trials were neural response latencies from the go cue that were similar to, or even longer than those during manual reaches (animal Y: MC activity peak 259.55 ± 31.58 ms versus BCI 255.05 ± 29.09 ms; animal Z: MC activity peak 353.05 ± 43.13 ms versus BCI 463.72 ± 43.86 ms; see S3I Fig).

## Motor planning activity in M1-PMd reflects a re-association strategy

To better understand how the FP network adjusts to altered feedback, we examined changes not just during the control of movement but also during its planning phase. To measure neural changes during planning, we employed an offline decoder approach similar to the one used for studying the movement phase. For planning, similar to movement decoding, we trained the decoder with baseline trials but used neural activity from the memory period. Unlike movement decoding, this training involved regressing the firing rates against a directional vector that continuously pointed to the cued target. Decoding the direction every 50 ms enabled us to reconstruct hypothetical 3D trajectories offline during the planning phase. For each target direction, we took the 400 ms time interval before the 'go' cue. We then calculated the vector sum of the individual unit vectors, each pointing in the direction decoded within that specific time bin (Fig 5A). The hypothetical trajectories we produced stem from a direct projection of the neural space into the 3D task space through the decoder, reflecting the intended direction during planning. In essence, these trajectories transform neural activity into a space where variations are explicitly linked to the task space (via the decoder), such as aiming at a target. To validate our decoding methodology during the planning phase, we identified the intended targets by determining the closest target to the final

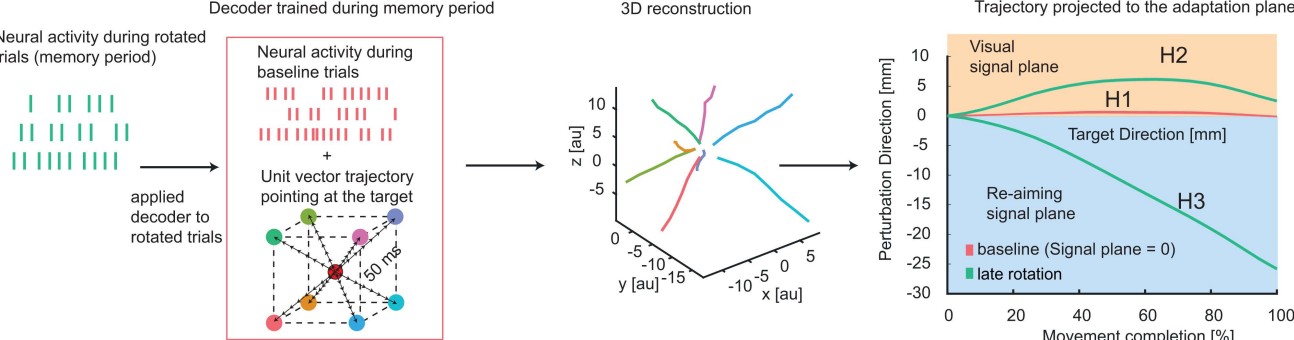

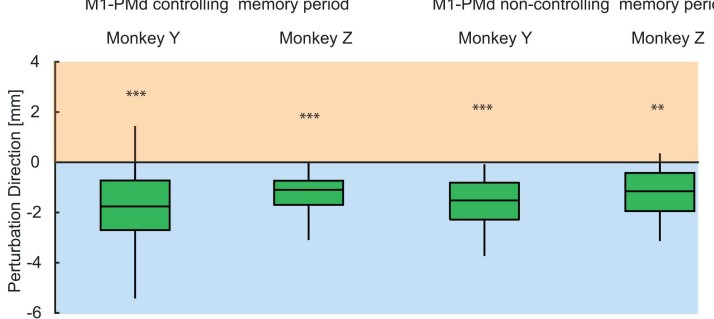

**Fig 5. Memory period decoder. (A)** Schematic illustration of the offline decoding principle during the memory period. As previously described for movement decoding, a velocity Kalman filter decoder was applied to reconstruct theoretical memory trajectories from neural activity (first column). During the memory period, a unity vector always pointing at the target was used for regression with neural firing rates (second column, training set). After calibration during baseline trials, the decoder was used to reconstruct continuous trajectories during perturbation trials (third column). The example averaged trajectories demonstrate continuous movements toward the target. In the last column, the reconstructed 3D trajectory from the offline decoder is projected onto the perturbation plane for perturbation trials. These projections assess visual-like or motor-like encoding, where positive deviations along this axis correspond to the visual signal, and negative deflections indicate motor output during adaptation. **(B)** The reconstructed trajectories in the perturbed dimension exhibit a negative deflection, indicating re-aiming, even during the planning period. At the end of the trajectory, a box plot compares all M1-PMd neurons with zero. The whiskers represent the 5th and 95th percentiles. Numerical data are available in S4 Data.

position attained at the conclusion of the planning phase for each reconstructed trajectory within the task space (using an 8-way discrete classifier). In our baseline trials, the decoder produced hypothetical trajectories in the task space that were consistent with the intended target (S5 Fig).

Equivalent to the analyses during movement, we tested the decoder's generalization to quantify adaptation and its spatial frame of reference. This was done by training it during the memory phase of the baseline trials and then applying it during the memory phase of the rotation trials. Much like neural activity observed during movement, any deviations in the reconstructed hypothetical trajectories (once projected onto the perturbation direction) would suggest either motor- or visual-like adaptation. We quantified this in a manner equivalent to how we assessed data during movement control (Fig 5A, right side). For both monkeys, the controlling and noncontrolling units in the M1-PMd regions exhibited motor-like adaptation during the planning phase. This demonstrates the use of a re-association strategy in a motor-reference frame during motor planning (Fig 5B). In the PRR, re-association in a motor-reference frame during planning could also be observed. Differences were statistically significant for the individual monkeys in either only the controlling units (monkey Y) or in the noncontrolling units (monkey Z) with nonsignificant trends in the respective other group of neurons (S6 Fig).

## Correlated changes in planning and movement-related activity strengthens over the course of adaptation in M1-PMd and PRR

By extracting the re-associated movement directions in both the planning and movement periods, we tested whether the observed level of trial-by-trial adaptation during planning predicts the adaptation levels during movement. We correlated the deviations of the decoded trajectories from a direct-to-target trajectory towards the end of the planning period (400 ms before go cue) with the deviations observed during movement 200 ms after the go cue for each trial (Fig 6A). We chose the early 200 ms deflection in the movement to reduce the impact of feedback-driven online corrections, which would be anticipated at later intervals. A positive correlation would support the hypothesis that adaptation is mostly achieved by adapting the initial conditions prior to the onset of movement, while movement-associated neural dynamics remain comparably stable. It's important to note that trajectories in the task space are theoretical during the planning phase and are decoded differently (see Materials and methods) than during the movement phase. Therefore, the regression slope between the planning and the movement trajectories is in arbitrary units. However, relative values in the slope across

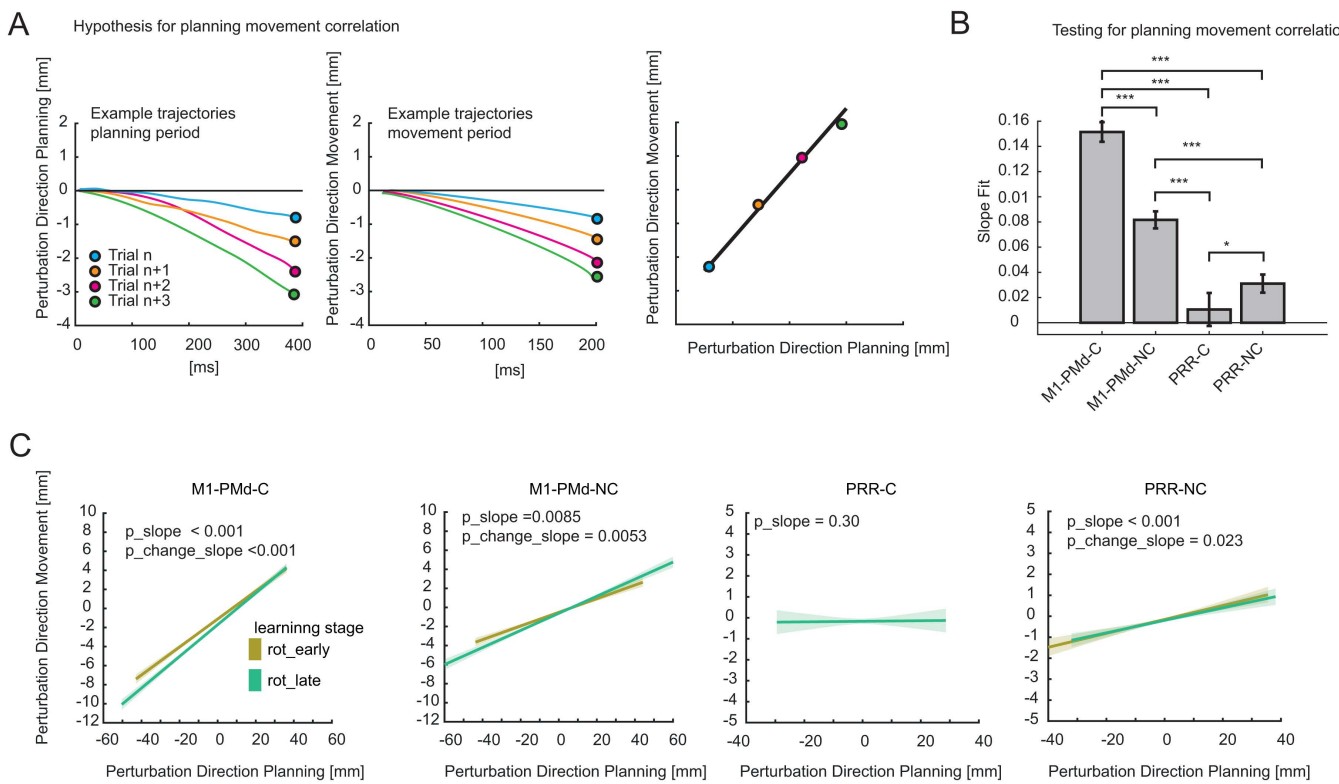

**Fig 6. Planning–movement relationship. (A)** Schematic representation of the analysis method. The reconstructed trajectories at the end of the planning period were regressed against those at 200 ms into the movement phase to test if adaptation levels during planning correlate with those during movement. For each axis, they represent the values assumed for each trial along the perturbation dimension in the specific period (planning and movement). **(B)** Multi-comparison of regression slopes for all controlling and noncontrolling populations. The slopes were extracted using a multiple linear regression model from data collected from two animals, with random factors considered. Paired multiple comparisons were deemed significant for an interaction effect between the two tested populations with a Bonferroni-corrected *p*-value < 0.05 (**p* < 0.05, ***\**p* < 0.001). The error bars indicate 95% Bonferroni-corrected bootstrapped confidence intervals as estimated by the model. **(C)** Graphical representation of learning-related changes in slope. As in the schematic in panel A (right), each plot shows, for every neural population, the average fitted line for the first half of the trials (yellow) and for the second half of the trials (green). Numerical data are available in S5 Data.

brain regions, different neuronal groups, or between the initial and advanced stages of adaptation can still signify the ratio at which changes related to adaptation during planning manifest as dynamic changes during movement.

We quantified the trial-to-trial relationship between deflections during planning and the direction of subsequent movement using a mixed-effects linear model. This model incorporates the trial number (perturbation trials) as an interaction factor to examine if the relationship evolves during adaptation. We applied a combined model, accounting for both animals, with the individual animal as a random factor and the day of the experiment nested within the animal random factor. Additionally, we separated the analysis for different brain areas and further categorized it based on controlling and noncontrolling units. Significant correlations were observed between deflections during planning and movement across all brain areas, but not all subgroups of neurons (Fig 6B). For controlling and noncontrolling M1-PMd populations, slopes were significant ($p < 0.001$ and $p = 0.0085$, respectively), indicating correlations. Similarly, for PRR noncontrolling units, a significant correlation was observed ($p < 0.001$). For PRR controlling units, such a relationship was not evident ($p = 0.30$). When we compared the slope values across all controlling and noncontrolling populations in the absence of the interaction parameter, we noticed a significant difference in every comparison (Fig 6B).

The linear relationship between planning and movement was already apparent during the baseline BCI trials and, though less pronounced, also during the MC trials (S7 Fig).

The fact that neural states during late planning and initial movement are correlated indicates that adaptation-induced changes during planning manifest as dynamic changes during movement. We next ask if this relationship remains unchanged over the course of learning, implying a stable relationship between initial state before movement and consecutive motor-related dynamics, or if the relationship evolves as learning progresses, implying a dynamic interplay changing the "gain" with which neural adaptations during planning are linked to changes in movement execution. The regression slope changes for both controlling and noncontrolling neural populations in PMd-M1, as well as in the noncontrolling population of PRR (Fig 6C shows the first half of the trials versus the second half of the trials). This change is indicated by a significant interaction of correlation strength with trial number in our model (for M1-PMd controlling $p\_change\_slope < 0.001$, for M1-PMd noncontrolling p_change_slope = 0.0053, for PRR noncontrolling $p\_change\_slope = 0.023$, for PRR controlling $p\_change\_slope = 0.4$. This suggests that, over time, the same degree of neural adaptation during the planning phase results in more pronounced motor corrections (interaction M1-PMd controlling: 0.000223, interaction M1-PMd noncontrolling: 0.00010. At the same time this translates in the opposite effect in PRR noncontrolling (interaction effect: −0.00010).

## Discussion

Previous studies on motor learning in primates have predominantly focused on the motor cortex in 2D settings [11,22,23,47,56,58]. Here, we investigated the adaptive mechanisms underlying short-term adaptation in a distributed network involving both frontal and parietal sensorimotor areas. By employing a VMR task in 3D movements, we explored motor adaptation and the associated neural dynamics with the same dimensionality as natural movements [69], reducing physical constraints. To minimize potential confounds arising from changes in cross-modal congruency due to the applied perturbation, we employed a BCI paradigm which removes movement-contingent proprioceptive feedback. By selectively manipulating the remaining task-relevant visual feedback and exclusively controlling movements through observed neurons feeding into the BCI decoder, we gained precise control over the sensorimotor loop and associated transformations [16,17]. Furthermore, we utilized memory-guided reaches to examine how motor planning activity and its relationship to movement are influenced by learning in both frontal and parietal sensorimotor regions. Our findings revealed coherent neural adaptation in a motor-reference frame across both frontal and parietal motor areas, i.e., including the parietal region remote from the frontal neurons controlling the BCI. This adaptation was observed in both planning- and movement-related activity, regardless of whether the neurons controlled BCI movements or not. Our results suggest the presence of a distributed mechanism of "re-association" which is happening in a motor-reference frame and influences initial states of movement associated neural dynamics during movement-planning already, both in frontal and parietal cortices.

## Distributed modulation of frontoparietal cortices under preserved network structure during VMR adaptation

A key finding of our study on BCI-VMR is that even neurons in the PRR area exhibit changes in neural activity during adaptation. Notably, this change is evident even though PRR neurons do not directly govern BCI movements and are remote from the controlling units in PMd-M1. By maintaining a consistent physical state (posture) during BCI movements, we ensured that proprioceptive input remained constant. Consequently, sensory changes induced by perturbations were confined to the visual cursor, which was directly and solely managed by the controlling units via the decoder [16]. Therefore, the decoder input represents the motor command, while its output corresponds to the task-relevant sensory feedback. Additionally, using offline decoding, we could analyze the encoding in groups of neurons not involved in online control.

Earlier research has highlighted the preservation of covariance and neural manifold structures during short-term BCI learning tasks in the motor cortex's controlling units [15,20]. We expand this notion by demonstrating that both noncontrolling motor cortex units and PRR units adapt, while preserving their correlation structure (S3 Fig). This within-manifold adaptation suggests that the constraints imposed by the underlying neural circuitry remains relatively unchanged during VMR learning [20,70,71]. Our findings indicate that a correlated network structure is maintained within the widespread frontoparietal sensorimotor cortices during fast BCI adaptation.

In this study, we chose to apply a perturbation in the fronto-parallel plane, as we previously found that subjects adapt most easily to visual perturbations of the cursor in this orientation. It remains an open question whether perturbations around other axes, which have been shown to be more difficult to learn in a human VMR study [67], might require more complex changes in neural activity, potentially extending beyond the intrinsic manifold. On the other hand, mixed representations of neural information about reach direction and distance (depth) in posterior parietal cortex [72–74] and also in PMd [75] do not suggest fundamental differences between dimensions.

## Consistent encoding of motor corrective variables in controlling and noncontrolling units during adaptation

Our findings indicate that noncontrolling M1-PMd and PRR units reflect adaptive changes in the same motor-related spatial frame of reference that controlling units use. In other words, the changes in noncontrolling units predominantly reflect the adapted movement directions that the controlling units have to produce to counteract the feedback perturbation. The uniform encoding of the movement-associated corrective variable across the frontoparietal network may facilitate efficient information exchange within a closed-loop system for motor corrections, especially involving direct PRR engagement [76]. Our results add to previous work in which parietal area 5 was found to encode motor errors, and where electrically stimulating this area generated movements opposite to the PD of the stimulated cells [77], in line with results from motor areas M1 and PMd [78]. The conclusion that PRR generates a signal in the same frame of reference as the motor areas, in our case, is based on the analysis of neural population responses during a motor adaptation task, without a potentially confounding co-varying proprioceptive feedback.

In tasks that impose a rule, requiring animals to learn and apply overt, context-specific stimulus-response associations, such as an anti-reach task [79], animals might be able to use an "explicit" strategy and solve context-specific spatial remapping of a target stimulus onto the motor goal by re-associating different neural dynamics with preexisting neural manifolds [36]. A closely related form of rule-based, explicit learning has been documented in discrete BCI experiments in monkeys [18] and humans [21]. Those studies employed a binary classifier to choose one of two targets, with the "cursor" shown only at the start and end of each trial, so continuous movement-contingent feedback was absent. Because the subject must discover and then apply a categorical association, the resulting adaptation may be based on an explicit strategy. The corresponding neural adaptation has been termed intrinsic-variable learning, i.e., re-association that reuses a preexisting neural manifold.

Our experiment differs in at least one key aspect. We used a continuous VMR paradigm in which the cursor was visible throughout the reach, providing a moment-by-moment error signal since the feedback deviated from the expected

feedback in perturbation trials. Under these circumstances, human psychophysics with classical VMR tasks often suggest primarily implicit recalibration of the visuomotor map [32,67,80,81]. Yet, learning in continuous VMR tasks may also involve a mixture of implicit recalibration and explicit re-aiming strategies [82]. Moreover, explicit processes themselves are not unitary but consist of dissociable strategies that interact with implicit recalibration in flexible ways [83]. The balance between these processes depends strongly on preparation time: explicit re-aiming requires longer preparation intervals, whereas restricting preparation to ~250 ms suppresses its use and yields behavior dominated by implicit adaptation [84,85]. Since our task design included a delay period prior to movement initiation, it may have provided the animals with additional opportunity to engage explicit strategies. Thus, while we interpret the re-association signals we observe across PMd, M1, and PRR as likely reflecting implicit recalibration, we cannot rule out contributions from explicit re-aiming. In either case, our findings reinforce that re-association is a general neural mechanism, expressed not only in rule-based explicit tasks but also in predominantly implicit adaptation contexts such as continuous VMR.

## Re-association signals during planning and movement control

In our study, we probed the trial-by-trial adaptation within the frontoparietal network during the movement planning phases. Specifically, we were keen to understand how re-association processes during planning are transferred to the execution of movement. Drawing from dynamical systems theory, we considered the possibility that neural activity during planning could establish the initial conditions, thereby guiding the dynamical system to produce adapted movements without the need for explicit cognitive computation [60] or enhanced restructuring of the network.

Our observations revealed linear correlations between trial-by-trial adjustments of planning and movement activity, evident in both controlling and noncontrolling units of frontal motor areas M1-PMd and, less pronounced, in noncontrolling units of PRR. From a dynamical systems perspective, this suggests that the adaptation of initial states to define updated starting conditions for the following dynamic evolution of states during movement might be particularly true for frontal, but less for parietal areas. Additionally, we found an enhancement of this relationship over the learning phase, indicating that changes during planning translated to more pronounced changes in movement in the later adaptation phase. This suggests that not only the motor error was reduced, but also that the adaptive mechanism became more efficient over the course of learning.

These findings partly challenge the notion of a linear dynamical system in motor learning, suggesting that the dynamical system itself may undergo changes. However, [86] offer an complementary perspective, proposing that evolving relationships may reflect the brain's effort to optimize its energetic costs. In this model, efficient control strategies favor selectively targeting dimensions that most impact future motor outcomes, reducing redundant preparatory effort. Supporting this hypothesis, we observed that as learning progressed, adjustments in planning became more effective in driving corrective changes in movement. This shift implies that less extensive re-association during planning could still enable more substantial corrections during movement, aligning with the view that learning serves not only to reduce error but also to refine the efficiency of neural control mechanisms in an energetically optimal way. We conducted a comparative analysis of online and offline decoding, revealing that all neurons, including noncontrolling units, coherently encode a corrective variable during motor control to counteract visuomotor rotations. This re-association mechanism, previously observed in motor areas, supports efficient information transfer in a feedback control-loop [11,15,18,21,28]. Additionally, our results indicate that re-association during planning can predict movement correction, although their relationship varies throughout learning and depends on the area and causal relationship between the area and the BCI movement. During the later stages of adaptation, it appears that less re-association is required during planning to achieve a specific degree of movement adaptation.

In conclusion, our study demonstrates that generalized adaptation mechanisms operate within a distributed neural network during three-dimensional movements, offering insights into neural manifold structures and motor error correction. By employing a VMR task in a 3D workspace and using a BCI, we captured spatial encoding and motor adaptation without physically constraining movements and independently controlling sensorimotor transformations from proprioceptive feedback.

Our findings highlight the critical role of an integrated motor and parietal sensorimotor network in adaptation. Both frontal and parietal regions primarily encode adapted motor commands rather than merely responding to perturbed visual feedback, supporting the re-association hypothesis of VMR learning. Moreover, learning-associated changes occurred not only in neurons controlling the BCI output but also in neurons not directly connected to the BCI, indicating a widespread mechanism of motor adaptation.

Finally, neural adaptation manifests in both planning and movement-related activities across the brain, with learning-associated changes in the correlation between these phases. This relationship evolved particularly within frontal areas during learning, suggesting different adaptation trajectories in frontal versus parietal regions. Our results underscore the distributed nature of adaptation within neural networks underlying motor control and highlight the importance of this network in fine-tuning spatial encoding for precise movement correction.

## Materials and methods

### Ethics statement

All procedures and experiments were authorized by the relevant regional authority (Niedersächsisches Landesamt für Verbraucherschutz und Lebensmittelsicherheit [LAVES]) under the permit numbers 3,392,42502-04-13/1100 and 33.19-42502-04-18/2823, and fully conformed to German law and the European Directive 2010/63/EU governing the use of animals in scientific research.

### Stereoscopic 3D virtual reality setup

The setup allowed the animals to perform reaching movements in a three-dimensional VR environment [67,87]. The animals were seated in a dark room with two monitors (BenQ XL2720T, 27-inch diagonal, 1920 × 1080 px, 60 Hz refresh rate) positioned on either side (Fig 1A). They looked through two semi-transparent mirrors (75 × 75 mm, 70R/30T, Edmund Optics) angled at 45° relative to the monitors. This created a stereoscopic 3D virtual impression in front of the subjects. The monitors were tilted 30° relative to the horizontal plane so that the workspace was projected below eye level, allowing the animals to perform ergonomic arm movements in front of their body.

To optimize the perception of the virtual workspace, the interpupillary distance of the monkeys was measured (Monkey Y = 33.2 mm, Monkey Z = 33.1 mm). The setup software and screen projections were then calibrated to these individual values to minimize discrepancies between the images for the left and right eyes. Eye position was tracked at 2 kHz (Eye-Link 1000 Plus, SR Research LTD, Ottawa, Canada), and the gaze was required to be maintained on the central fixation point during the memory period (see below).

### Manual and BCI control software

The custom-written task controller was implemented in C++, allowing simultaneous tracking of multiple interfaces including the BCI, the eye and the hand tracking systems [88]. The BCI was implemented as a Matlab program running an online loop, which communicated with the task controller via a Matlab implementation of the VRPN library through a MEX file [89]. Neural data were not just recorded to hard drives but also processed online through a MEX interface provided by the recording system (Cerebus, Blackrock Microsystems, Salt Lake City, USA) to extract spike counts (see below). The loop operated every 50 ms in consecutive nonoverlapping time windows. During BCI trials, the cursor was automatically placed at the central fixation point during the holding phase and after the movement was completed (see task description below).

### Tracking system calibration

Four infrared cameras tracked the position of four passive markers centered on the monkey's palm in real-time (Fig 1A) at 100 Hz. The two animals were trained to wear a 3D-printed plastic framework containing the four reflective markers

arranged around the right wrist. The center of geometry of the four markers was tracked using four Vicon Bonita B10 cameras (Vicon, Oxford, UK) and aligned with the midpoint of the palm. To facilitate accurate tracking and control in the VR setup, we performed two calibrations. First, we calibrated the cameras to stream the center of geometry of the markers attached to the monkey's wrist. We added an extra marker at the position of the monkey's wrist, which served as the reference point during this calibration. Using Vicon Tracker 1.3.1 software (Vicon), we adjusted the center of geometry of the four markers to align with this additional marker. This allowed the software to stream the position of the center of geometry online to the task controller, ensuring that this position corresponded accurately to the monkey's wrist position in the physical setup.

Second, we calibrated the camera space with the task controller space to ensure proper alignment between the virtual environment and the physical setup. During this calibration, a human operator co-registered the cursor position and the displayed target positions using semi-transparent mirrors. This process translated the coordinates from Vicon space to the task space, allowing the VR system to accurately reflect the monkey's movements within the task environment.[67].

### Center-out reach task

Two Rhesus monkeys performed a 3D, memory-guided, and center-out reach task (Fig 1B). At the beginning of each trial, during the decoder calibration phase, the animals were required to acquire and maintain a central fixation point with both their hand and gaze for 400 ms (fixation period). Following this, one of eight targets, positioned at the vertices of a 70 mm-sided cube and centered around the fixation point, was briefly displayed for 300 ms (cue period). The animals had to continue holding their hand and gaze at the center of the cube for an additional 400 ms (planning period), followed by a variable delay of up to 600 ms. After the delay, the central fixation target disappeared, signaling the animals to move toward the target (movement period), with a timeout limit of 1,500 ms.

During BCI control trials, the cursor movement was controlled by a neural decoder. The task remained identical to the MC task that had been used for calibration. If, at any point, the animals directed their gaze outside a 30 mm tolerance window around the fixation point or moved their hand more than 15 mm from its starting position (security radius) during the fixation period, the trial was aborted. MC trials were employed to calibrate a biomimetic velocity KF decoder, which was subsequently used to perform the BCI trials (Fig 1C, BCI).

Once the animals became proficient in controlling the cursor during the BCI task (reaching a baseline of 160 successful trials), we introduced a VMR to perturb the visual cursor feedback. This was implemented by applying a 30° visuomotor rotation of the cursor's movement in the fronto-parallel plane. The direction of the rotation (clockwise or counterclockwise) varied between days, and the perturbation lasted for a variable number of trials with a maximum of 320 trials. Following the perturbation phase, a washout phase was introduced, during which the visuomotor rotation was removed, and the animals returned to controlling the cursor without any rotation for another variable number of trials until they finished their session for the day.

Both animals were housed socially with a single male of the same species at the German Primate Center. The enclosure sizes surpassed the standards required by both German and European regulations. They were kept in a setting enriched with wooden structures, toys, and various other enrichment devices.

### Neural recordings

Neuronal activity was recorded from two macaque monkeys implanted with floating microelectrode arrays (FMAs, Micro-Probes, Gaithersburg, USA) in three cortical areas: M1, PMd, and PRR. Monkey Y was implanted with two 32-channel FMAs in each area, while monkey Z was implanted with three 32-channel FMAs in PMd and PRR (see S1C Fig). Both animals were implanted in the left hemisphere, contralateral to the arm used for calibrating the BCI decoder and performing the MC task (S1C Fig). Premotor array positions in monkey Z range between 18.2–25.2 mm AP and 10.5–13.5 mm ML

(array centers), and in monkey Y between 18.6–19.6 mm AP and 13.1–17.4 mm ML. M1 array positions in monkey Z range between 14.2–17.2 mm AP and 12.5–14.5 mm ML, and in monkey Y between 10.3–11.6 mm AP 11.9–16.2 mm ML. Parietal array positions in monkey Z range between −1.3 and −6.8 mm AP and 5.5–10.5 mm ML, and in monkey Y between −0.4 and −4.1 mm AP and 8.1–11.9 mm ML. Position data per each array can be found in S1 Fig. The positioning of the premotor arrays aimed at the arm region of PMd. M1 arrays targeted the arm region along different depth in the anterior wall of the central sulcus. Parietal arrays covered the MIP region of the posterior PRR with the longer electrodes, while the shorter electrodes sample from closer to the surface and may partially include area 5 (also referred to as area PEc).

In single-day sessions, we recorded from 128 electrodes simultaneously in monkey Y, while we recorded from all 256 electrodes in monkey Z. Activity was recorded online and stored on a hard drive using a 128-channel Cerebus system (Blackrock Microsystems, Salt Lake City, USA) for monkey Y and two 128-channel Cerebus systems for monkey Z. Data were sampled at 30 kHz. Signals were band-pass filtered (250 Hz–7.5 kHz), and a threshold of −4.5 times the root-mean-square (RMS) voltage was used to isolate below-threshold waveforms.

Before the start of each daily experimental session, manual online sorting was performed by an expert user. Sorting involved identification of clusters in PC space to classify neuronal activity based on waveform characteristics.

### Neural decoder

Online neuronal activity during movement was used to train a velocity KF decoder [90] by regressing neuronal firing rates from a subset of single and multi-units with hand velocity (50 ms steps, nonoverlapping time windows) while the monkey manually performed the task during the movement period. During FO decoding a subset of M1 and PMd units where used while during FP decoding subsets of M1 and PMd units were combined with all PRR units. After an initial MC calibration consisting of about 60 trials for initializing KF parameters, the velocity KF decoder was retrained using velocity vectors that directly pointed toward the target [91]. During retraining, first the computer mostly controlled the cursor movement. The computer's contribution then was step-by-step progressively reduced until real-time neural activity fully guided the cursor. The transition was managed by applying a weighted vectorial sum of both the computer-generated and neural signals, gradually increasing the brain's control for smoother cursor operation. This adjustment was performed over successive blocks of 20–30 trials, reducing the computer's contribution from 70% to 0%. The process was not automated and required expert judgment to determine the appropriate timing of the transitions. Additionally, hand movements greater than 15 mm from the resting position were discouraged to prevent the animal from using its hand during BCI trials. This retraining calibration step was necessary to eliminate hand movement during BCI control.

### Projection of trajectories on the adaptive space

We employed the method described by [28] to average trajectories reaching different targets. In brief, trajectories were first projected onto the plane of the applied perturbation (xy—fronto-parallel plane). For each trajectory to a target, the dimension orthogonal to the direction from the center to the specific target was defined as the "perturbation dimension." This allowed all trajectories to the eight different targets to be averaged within this common reference frame. The positive sign of this second axis was chosen based on the direction of the applied perturbation (clockwise or counterclockwise, depending on the session).

### Neural data pre-processing (offline)

The online-sorted spikes were re-sorted offline using Boss software (v.1.0.3, Blackrock Microsystems, Salt Lake City, USA) to better isolate and identify additional units among the noncontrolling units. Units used for online decoding were left unchanged. A threshold of −4.5 times the standard deviation of the noise was used to isolate spikes, with a refractory period of 1.5 ms. Clusters were identified using a semi-automated k-means method in PC space. This semi-automated process required the user to specify the number of clusters and define the initial centroid for each.

## Offline trajectory reconstruction

To study changes in the encoded correction variable during adaptation for noncontrolling units, and during the memory period for all units, we applied a KF decoder offline, similar to the one used for online BCI trials, to reconstruct cursor trajectories. The decoder was always trained on the activity from baseline trials and then applied to the activity from the rotated trials to identify the encoded variable during adaptation.

For the evaluation of the decoder during baseline trials, a leave-one-out cross-validation procedure was used to include all baseline trials in the analysis. Each time point, sampled every 50 ms and containing all the spikes from each neuron, was used as input to the decoder. For the decoder's training output, real movement speeds were used during the movement epoch (sampled every 50 ms), while a normalized vector pointing to the cued target was used during the planning epochs (also sampled every 50 ms). This approach allowed us to reconstruct theoretical neural trajectories from the decoded speeds during movement for noncontrolling units and during the memory phase for both controlling and noncontrolling units.

## Manifold and alignment index

We calculated the PC space for each analyzed epoch and region, plotting the variance explained by each dimension (S3 Fig), averaged across the dataset. In the cross-projected analysis of S3A, S3C, S3E, and S3G, we computed the PCs during the movement phase, projected the activity from the memory phase onto the seven PCs, and calculated the explained variance in each specific direction (component). For the alignment index calculation in S3B, S3D, S3F, and S3H Fig, we followed the method of Elsayed and colleagues [62], considering the top four PCs in the transformation.

In brief, the alignment index measures manifold alignment and is calculated as the ratio between the explained variance of the projected activity (e.g., during the planning phase) along the original dimensions and the explained variance of the original main dimensions (e.g., derived from movement activity). To calculate the original PC space, we used cross-validated baseline trials to ensure that the alignment index reflects genuine differences between baseline and experimental phases, avoiding artifacts from overfitting.

## Decoder contribution

Neuronal contributions were computed from the observation matrix, the rows of which contain the weights linking $x$, $y$, $z$ speed to the expected firing rate of neuron. We determined the firing rate of a single cell given the direction of movement using the following equation:

$$FR_t = b_0 + b_1 V_{xt} + b_2 V_{yt} + b_3 V_{zt}$$

We calculated the amplitude of the regressing vector ($b_1$, $b_2$, $b_3$), summed to $b_0$, to estimate the single neuron's contribution to the decoder, as shown in Fig 4A. This measure provides a single scalar that reflects how much leverage the neuron can exert on the decoded cursor state.

## Generalized linear model

We used a linear model to test the significance of the relationship between the planning and movement phases in brain each area, separately for the groups of controlling and noncontrolling units. We also included trial number as variable to model slope changes due to learning and tested the interaction between learning and the planning-movement slope. Different experimental sessions and the two animals were treated as random factors in the model, with a nested hierarchical structure. The Wilkinson notation for the modeled variable was as follows:

$$\text{Re–aiming\_movement} \sim 1 + \text{Re–aiming–memory} + \text{Re–aiming–memory} : \text{trial\_number} + (1|\text{animal/recording\_session})$$

To calculate slope differences between controlling and noncontrolling populations, as well as between different areas, we employed a similar model that incorporated pairwise-tested interaction effects for the areas being compared. All six pairwise comparisons were corrected for significance using the Bonferroni method (see Fig 5B). The corresponding Wilkinson notation was as follows:

$$\text{Re-aiming\_movement} \sim 1 + \text{Re-aiming-memory} + \text{Re-aiming-memory} : \text{area} + (1|\text{animal/recording\_session})$$

## Supporting information

**S1 Table.    Experimental design.** This table summarizes, for each recording session, the number of units for each population, the type of decoder used, the rotation angle, and the direction of rotation (clockwise, CW, or counterclockwise, CCW).
(PDF)

**S1 Fig.    Hand movement speed and array implantations. (A)** VR setup used for concurrent manual control (MC) and brain–computer interface (BCI) trials. **(B)** Comparison of real hand movement speed during BCI trials (cyan) and cursor speed during BCI trials (pink). The graph displays the speed of hand movements executed by the monkey during BCI trials as well as the cursor speed during BCI trials, along with the Pearson correlation coefficient indicating the very low correlation between these two speeds. Numerical data are available in S6 Data. **(C)** Array locations. Monkey Y received six 32-channel floating micro-wire arrays (FMA, MicroProbes for Life Sciences, Gaithersburg, USA) in three distinct brain regions contralateral to the arm used for decoder calibration. Two arrays with staggered electrode lengths ranging from 7.1 to 1.9 mm were implanted in the arm region of the primary motor cortex (M1), two with staggered electrode lengths ranging from 4.5 to 2.1 mm in the dorsal premotor cortex (PMd), and two with staggered electrode lengths ranging from 7.1 to 1.9 mm in the parietal reach region (PRR). Similarly, in Monkey Z, three arrays were implanted in PMd, two in M1, and three in PRR. Stereotactic coordinates (AP, ML) in mm along the anterior-posterior and medio-lateral axes are provided in the sketches for each array. All the pictures in the figure were taken by the authors.
(EPS)

**S2 Fig.    Alignment index during task epochs.** Alignment indexes (AI) during various task epochs for planning (left) and movement (right) phases were calculated. The AI was determined by projecting the neural activity from both the early and late adaptation phases onto the principal components (PCs) derived from baseline activity. We selected a sufficient number of PCs to explain 90% of the baseline variance. AI during the baseline was computed using cross-validation. Notably, AI exhibited minimal variation between the baseline and visuomotor rotation (VMR) phases, indicating the preservation of neural manifolds throughout the learning process. Numerical data are available in S7 Data.
(EPS)

**S3 Fig.    Neural manifold comparison between MC and BCI. (A)** Explained variance: Monkey Y's neural activity during MC planning was projected onto principal components (PCs) calculated from neural activity during movement. The explained variance for both activities used to calculate the components (movement training, green) and for cross-validated trials not used in component calculation (movement test, blue) is shown. Bars show 95% confidence intervals. **(B)** Alignment index (Monkey Y – MC): alignment index for Monkey Y calculated during MC for M1-PMd (left) and PRR (right). It measures the extent of overlap among neural manifolds during planning and movement execution using the first four PCs. The whiskers represent the 5th and 95th percentiles. **(C)** Explained variance (Monkey Y – BCI): similar to A, but for BCI trials. **(D)** Alignment index (Monkey Y – BCI): similar to B, but for BCI trials. **(E)** Explained variance (Monkey Z): same analysis as A, but for Monkey Z. **(F)** Alignment index (Monkey Z): same analysis as B, but for Monkey Z. **(G)** Explained variance (Monkey Z – BCI): same as C, but for Monkey Z. Numerical data are available in S8 Data. **(H)** Alignment index

(Monkey Z – BCI): same as D, but for Monkey Z. **(I)** Population-averaged firing rates as a function of time aligned to the go-cue. The plots for the two animals, showing the mean and confidence intervals, illustrate the time-varying neural similarities between manual and BCI control.
(EPS)

**S4 Fig. Population tuning shifts as a result of adaptation. (A)** Average shift of the preferred direction (PD) during the movement period in late rotation trials. For each neuron, the PD shift was computed between rotation and baseline trials. Averages were computed separately for controlling and noncontrolling neurons in areas M1 and PMd, respectively. Error bars show standard errors of the mean. **(B)** Same as in (A), but during the planning period. Numerical data are available in S9 Data.
(EPS)

**S5 Fig. Memory period decoder. (A)** For Monkey Y, averaged trajectories of the last 50% of trials reconstructed with the memory period decoder, similar to Fig 5, but for PRR controlling and noncontrolling units. **(B)** Same as in (A), but for Monkey Z. Numerical data are available in S10 Data.
(EPS)

**S6 Fig. Target classifier from memory period decoder.** Using the method described in main Fig 5, we accurately classified the cued target with performance significantly above chance level (12.5%, corresponding to one correct choice out of eight possible predefined targets). This classification was based on analyzing the trajectory endpoints within the spatial framework at the end of the memory period. By examining where the trajectory endpoint was located, we were able to infer which of the eight predefined targets had been cued. Numerical data are available in S4 Data.
(EPS)

**S7 Fig. Planning—movement relation during baseline BCI and manual control trials.** Similar to the planning–movement relationship observed during adaptation (main Fig 6), a linear relationship between deflections in the planning and the movement period can be observed during the baseline manual control trials and BCI trials. Numerical data are available in S11 Data.
(EPS)

**S1 Data. Numerical data for Fig 2.**
(ZIP)

**S2 Data. Numerical data for Figs 3 and 4.**
(ZIP)

**S3 Data. Numerical data for Fig 4.**
(ZIP)

**S4 Data. Numerical data for Fig 5 and S6 Fig.**
(ZIP)

**S5 Data. Numerical data for Fig 6.**
(ZIP)

**S6 Data. Numerical data for S1 Fig.**
(ZIP)

**S7 Data. Numerical data for S2 and S3 Figs.**
(ZIP)

**S8 Data. Numerical data for S3 Fig.**
(ZIP)

**S9 Data. Numerical data for S4 Fig.**
(ZIP)

**S10 Data. Numerical data for S5 Fig.**
(ZIP)

**S11 Data. Numerical data for S7 Fig.**
(ZIP)

## Author contributions

**Conceptualization:** Enrico Ferrea, Alexander Gail.

**Data curation:** Enrico Ferrea.

**Formal analysis:** Enrico Ferrea.

**Funding acquisition:** Alexander Gail.

**Investigation:** Enrico Ferrea, Alexander Gail.

**Methodology:** Enrico Ferrea, Pierre Morel.

**Project administration:** Alexander Gail.

**Resources:** Alexander Gail.

**Software:** Enrico Ferrea, Pierre Morel.

**Supervision:** Pierre Morel, Alexander Gail.

**Validation:** Enrico Ferrea, Alexander Gail.

**Visualization:** Enrico Ferrea.

**Writing – original draft:** Enrico Ferrea, Alexander Gail.

**Writing – review & editing:** Enrico Ferrea, Pierre Morel, Alexander Gail.

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
