## [Editor Report · Decision Letter 0]

11 Dec 2024

Dear Dr Ferrea,

Thank you for submitting your manuscript entitled "Large-scale co-adaptation of frontal and parietal planning signals when learning to control a brain-computer interface" for consideration as a Research Article by PLOS Biology.

Your manuscript has now been evaluated by the PLOS Biology editorial staff and I am writing to let you know that we would like to send your submission out for external peer review.

Once your full submission is complete, your paper will undergo a series of checks in preparation for peer review. After your manuscript has passed the checks it will be sent out for review. To provide the metadata for your submission, please Login to Editorial Manager (https://www.editorialmanager.com/pbiology) within two working days, i.e. by Dec 13 2024 11:59PM.

Kind regards,

Christian

Christian Schnell, PhD

Senior Editor

PLOS Biology

cschnell@plos.org

---

## [Decision Letter · Decision Letter 1]

10 Feb 2025

Dear Dr Ferrea,

Thank you for your patience while your manuscript "Large-scale co-adaptation of frontal and parietal planning signals when learning to control a brain-computer interface" was peer-reviewed at PLOS Biology. It has now been evaluated by the PLOS Biology editors, an Academic Editor with relevant expertise, and by several independent reviewers.

In light of the reviews, which you will find at the end of this email, we would like to invite you to revise the work to thoroughly address the reviewers' reports.

As you will see below, the reviewers think that the study is interesting, overall well executed and provides important insights. Reviewer 1 mainly asks for additional methodological details, a few more analyses of the existing data and some textual clarifications. Reviewer 2 also asks for a few additional analyses to rule out alternative explanations and textual revisions, for example to flesh out the hypotheses better, clarify the novelty and the limitations of the task the monkeys were trained on. Another concern that we would like to highlight regards the imbalance in data between the two monkeys. In several cases, Monkey Z appears to have very few data points (for example, one t-test has df = 4, and another has df = 9), raising questions about the robustness of the data analysis and the strength of the overall conclusions, given that the study is based on only two monkeys.

Given the extent of revision needed, we cannot make a decision about publication until we have seen the revised manuscript and your response to the reviewers' comments. Your revised manuscript is likely to be sent for further evaluation by all or a subset of the reviewers.

**IMPORTANT - SUBMITTING YOUR REVISION**

*Re-submission Checklist*

*Published Peer Review*

*PLOS Data Policy*

*Blot and Gel Data Policy*

Sincerely,

Christian

Christian Schnell, PhD

Senior Editor

PLOS Biology

cschnell@plos.org

REVIEWS:

Reviewer #1: In this study, Gail's lab investigated the frame of reference for motor planning and execution in reaching using a brain-computer interface (BCI) paradigm. The study involved the primary motor cortex (M1), dorsal premotor cortex (PMd), and the parietal reach region (PRR).

The main finding is that both frontal and parietal regions reflected adapted motor commands more effectively than perturbed visual feedback during both movement preparation and execution. PRR exhibited motor-like encoding during learning, even when it was not directly controlling the movement, suggesting that PRR plays a more integral role in motor adaptation than merely reflecting visual feedback. However, the transfer from planning to movement was stronger in the frontal cortex compared to PRR.

This manuscript clearly demonstrates that both the frontal and parietal cortices encode reaching in a motor-reference frame, indicating a re-aiming mechanism for both cortices. This process occurs during both the planning and execution phases of reaching. The manuscript is well-structured, with clear hypotheses and well-designed methods to test them. The presentation is highly effective, with well-crafted figures that guide the reader through the reasoning (e.g., Figure 3).

I particularly appreciated the paradigm, which studies reaching movements in 3D rather than the conventional center-out task. In this paradigm, the BCI approach ensures constant proprioceptive feedback, making the task reliant solely on visual and motor signals. While prior work by the Batista lab demonstrated the preservation of the neural manifold in the M1-PMd cortex during short-term adaptation, this manuscript extends that concept to include BCI non-controlling units and PRR. This work is thus significant in extending the understanding of motor adaptation to both the parietal cortex and frontal neurons not driving the BCI.

Major Points:

1. M1 and PMd differentiation: It would be valuable to separate the neuronal pools of M1 and PMd and verify whether the results hold true for each region independently.

2. PRR targeting: Please clarify which specific area of PRR was targeted, considering the depth of the recording electrodes. For Monkey Z, in the supplementary material, include the position of the interhemispheric midline to better evaluate which portion of the posterior parietal cortex (PPC) was studied.

3. Frame of reference for reaching: On line 84, it is stated that "Preparatory activities include reach-goal information in different body-related frames of reference, including the predominantly gaze-centered parietal reach region (PRR)." Since different regions within the PRR area have been shown to employ various reference frames for reaching, this alternative version should also be mentioned (e.g., Bosco et al., 2016, Scientific Reports; Piserchia et al., 2017, Cerebral Cortex; Hadjidimitrakis et al., 2020, Journal of Comparative Neurology).

4. Discussion - 3D reaches: At the beginning of the discussion, the authors highlight the importance of studying 3D reaches rather than 2D movements. It would be appropriate to cite prior work from other groups that studied 3D reaches (e.g., Fattori lab in PPC, Filippini et al., 2020, Journal of Neural Engineering; Hadjidimitrakis et al., 2022, Cell Reports).

5. PRR decoding of motor plans: Regarding lines 290-300, some clarifications are needed:

o How was the low yield in PRR units mitigated?

o What does the statement "the controlling PRR neurons were combined with a subset of M1-PMd units in the FP decoder sessions" mean? Why was this necessary?

o How many units were recorded in PRR, and what was the threshold needed for statistical significance? What were the yields for M1 and PMd?

o The fact that in PRR we see similar "motor" characteristics may be it caused by the M1-PMd contamination (FP decoder session M1-PMd units added to PRR, lines 289-294)?

6. Working space limitation: The 3D reaching cube was 7 cm on each side, which does not encompass most of the working space. Was this small size chosen due to limitations in covering the peripersonal space? Please provide some insight and include a comment in the manuscript about the implications of this limited working space on the generalizability of the conclusions.

7. Perturbations: In lines 177-180, please justify why perturbations were not applied in the Z dimension (depth).

8. Results (lines 279-280): If I understand correctly, the non-controlling neural population adapts to the VMR as much as the controlling population for Monkey Z (94%, with 100% indicating the same level of adaptation). However, for Monkey Y, this measure drops to 59%. This discrepancy suggests a significant difference in the adaptation process between the two animals. Could the authors compute a null distribution or reference values? What is their interpretation of this difference?

9. BCI unit selection: The authors selected subsets of neurons to drive the BCI, using either frontal-only (FO) or mixed frontal-parietal (FP) units. While I assume this was done for performance reasons or due to limited motor coding in PRR, this rationale should be clearly stated. Lines 289-292 seem to address this but are somewhat unclear.

Minor Points:

1. Introduction (lines 83-91): This section seems confusing, possibly due to typos from older manuscript versions.

2. Results (lines 337-339): Include numerical results (or graphs) in the main text to support these claims.

3. Figure S5 legend: Add more detailed information to help readers better understand the data presented.

Reviewer #2: Ferra and colleagues present new BCI experiments to study how frontal and parietal motor areas contribute to short term decoder perturbations. Their study incorporates three new aspects that allow them to expand upon prior work examining adaptation to rotational-perturbations in BCIs: 1) recording from parietal regions, 2) comparing adaptation in the neural populations that drive the BCI ("controlling") and those that do not ("non-controlling"), and 3) using a delayed reaching task to examine planning activity. The paper generally presents clear data that support the hypothesis that frontal and parietal areas both contribute to adaptation mechanisms, and that neural activity patterns are consistent with a "re-aiming" strategy that occurs during planning phases. The work includes many challenging experiments and thorough analyses that are to be commended, and contributes to an active literature discussing learning mechanisms in BCI.

I have a few more major concerns:

1) The paper is currently written to test core predictions of the hypothesis that rapid visuomotor rotations are learned via "re-aiming" in BCI. This hypothesis is a predominant view within the many recent BCI learning studies. While further supporting established hypotheses is not a weakness fundamentally, the paper may benefit from framing the contributions to better center the novel contributions of the work.

2) An inter-related comment: The manipulations performed to allow the researchers to analyze planning activity (introducing a delay period task) may influence their findings confirming a "re-aiming" hypothesis. Past work in human psychophysics highlights a relationship between the time people have to prepare and the expression of "explicit" motor learning strategies such as re-aiming (e.g., Haith et al., J Neurosci 2015; Leow et al., J Neurophysiol 2017). Indeed, Leow et al. 2017 uses delay-time manipulations to isolate implicit vs. explicit contributions to visuomotor adaptation. By introducing a long delay period, the task design may increase the animals' ability to/likelihood to use explicit strategies like re-aiming. This does not refute the paper's claims, but does add important nuance to the interpretation of the results. In particular, the likely link between the task design and learning mechanisms, in my opinion, reduces the impact of the author's claims to test their central hypothesis about re-aiming.

3) One of the authors' notable (and interesting) claims is that planning activity and motor activity become more correlated with each other with learning. However, the presented analyses make it a bit difficult to fully appreciate and understand this effect. The analyses of planning and movement relationships are entirely restricted to the linear interaction model and a bar graph of coefficients. I'm a bit unclear on why, for instance, a plot similar to Fig S6 isn't included for the BCI data. The inclusion of the many possible contributing factors and interaction terms also makes it a bit challenging to interpret. I also struggle to fully understand the concluding sentence of that section ("This suggests that, over time, the same degree of re-aiming during the planning phase results in more pronounced motor corrections (data not shown)."

Specific modest/minor comments:

The paper neglects to include the N for statistical tests. Please update to include these.

The methods presented to describe how the authors analyzed each neuron's contribution to the BCI is insufficient. The authors say the used the "matrix from the Kalman Filter" but the Kalman Filter has multiple possible matrices that could be used for these calculations (the Kalman Gain matrix or the Observation model).

It's unclear to me that the authors conclusively rule out the possibility that the animals are trying to move during the planning phase with the current analyses. The current analyses show that the planning and movement PCs are not aligned during arm movements, and they see the same pattern during BCI. But there are aspects of the task that differ between the planning and movement phase that may contribute to this misalignment, such as the animal being allowed to release fixation. That transition is the same between BCI and the arm movement task. If those task-related factors are the primary driver of the alignment change - as opposed to the transition from planning to movement - this test would not reveal whether or not the animals are attempting to move in BCI.

The discussion paragraph related to differences between rule-based learning in discrete BCIs and the learning observed in this task is a bit unclear. The exact distinction the authors aim to draw between "rule-based" and explicit/implicit is not clear. I find this section particularly confusing in light of my comments above about how even "continuous" VMR tasks can use a mixture of mechanisms depending on the task structure.

I am not clear I understand what aspects of their findings the authors feel support the claims that: "…the adaptive mechanism became more efficient over the course of learning" and "During the later stages of adaptation, it appears that less re-aiming is required during planning…" (discussion)

In the intro: "fast adaptation, which is particularly desirable for BCI learning". Desirable by who? For BCI applications? What exactly do the authors mean by this? It is unclear what link the authors see between adaptation to artificially-introduced perturbations and learning that would occur in clinical BCIs, for instance.

The authors use the term "co-adaptation" to refer to learning mechanisms that seem to co-occur. Within the BCI field, the term "co-adaptation" is often used to refer to learning of the decoder and the brain (e.g., Taylor et al., Science 2003). The authors may benefit from choosing a term that is not already "loaded" with preconceptions for certain audiences.

Line 260: "As a result," phrasing is a bit confusing. Do you mean to highlight that this is a result you found?

The paper cites Jiang et al., 2020 within the manuscript but this is not included in the bibliography. This omission is particularly important since you're referring to a method, which I couldn't evaluate.

I would suggest that the statistical analyses/results presented in lines 305 - 319 could likely be presented much more concisely/efficiently.

The discussion states "This within-manifold adaptation suggests that functional connectivity remains relatively unchanged during VMR learning." None of the cited papers, however, directly prove this claim. All experimental papers cited do not analyze/measure functional connectivity. The modeling paper does make predictions about what connectivity changes do/don't need to change, but one key observation of that paper is that there is more nuance to the interpretation of the in/out of manifold BCI learning and the link between network connections (since it could also be explained by feedback available to guide learning in the RNN). Please revise.

The supplemental table of experimental sessions could benefit from some formatting revisions to make it easier to read (e.g., the alignment is inconsistent across rows; it's not always clear what number goes with which column label due to alignment).

---

## [Decision Letter · Decision Letter 2]

11 Aug 2025

Dear Dr Ferrea,

Thank you for your patience while we considered your revised manuscript "Frontal and parietal planning signals adapt in a motor reference frame when learning to control a brain-computer interface" for publication as a Research Article at PLOS Biology. This revised version of your manuscript has been evaluated by the PLOS Biology editors, the Academic Editor and the original reviewers.

Based on the reviews and on our Academic Editor's assessment of your revision, we are likely to accept this manuscript for publication, provided you satisfactorily address the remaining points raised by the reviewers. Please also make sure to address the following data and other policy-related requests:

* We would like to suggest a different title to improve its accessibility for our broad audience: "Frontal and parietal planning signals encode adapted motor commands when learning to control a brain-computer interface"

* Please add the links to the funding agencies in the Financial Disclosure statement in the manuscript details.

* Please mention in the competing interests section that Alexander Gail is a member of PLOS Biology's editorial board.

* DATA POLICY:

Regardless of the method selected, please ensure that you provide the individual numerical values that underlie the summary data displayed in the following figure panels as they are essential for readers to assess your analysis and to reproduce it: 3C, 4BC, 5B, 6B, S2, S3 (all panels), S4AB and S5 (all panels).

* CODE POLICY

We expect to receive your revised manuscript within two weeks.

*Published Peer Review History*

*Press*

Sincerely,

Christian

Christian Schnell, PhD

Senior Editor

cschnell@plos.org

PLOS Biology

Reviewer remarks:

Reviewer #1: All my suggestions have been implemented graciously.

Reviewer #2: I thank the authors for their thoughtful revisions to the manuscript and responses to my comments. The paper is significantly improved and largely addresses my concerns. I have two lingering issues after reviewing the new manuscript:

1. I do not feel the current discussion section adequately addresses my comments about distinctions between implicit/explicit computations. I take the authors point (in the rebuttal) that the link between these notions and "re-aiming" as discussed in the monkey/BCI literature is not yet clear. Yet, I think many studies (and, importantly, readers) often implicitly make this tempting link between the two separate lines of research. The authors seem to be arguing that explicit strategies cannot occur in continuous VMR tasks, but plenty of human psychophysics studies (e.g., the ones I cited in my prior critique) do observe a mixture of explicit and implicit strategies in continuous VMR tasks. I think it's important that this nuance and potential ambiguity is explicitly addressed in the discussion (relevant paragraphs: "In tasks that impose a rule, …" and the following one, "Our experiment differs in at least one aspect…".

2. The new manuscript includes the sentence "Instead what we observed where cursor movements and associated neural response latencies similar to manual reaches." I could not find any data provided to support this claim. Can you please clarify?

---

## [Editor Report · Decision Letter 3]

11 Sep 2025

Dear Enrico,

Thank you for the submission of your revised Research Article "Frontal and parietal planning signals encode adapted motor commands when learning to control a brain-computer interface" for publication in PLOS Biology. On behalf of my colleagues and the Academic Editor, Roi Cohen Kadosh, I am pleased to say that we can in principle accept your manuscript for publication, provided you address any remaining formatting and reporting issues. These will be detailed in an email you should receive within 2-3 business days from our colleagues in the journal operations team; no action is required from you until then. Please note that we will not be able to formally accept your manuscript and schedule it for publication until you have completed any requested changes.

PRESS

Sincerely, 

Christian

Christian Schnell, PhD

Senior Editor

PLOS Biology

cschnell@plos.org